# Glycan-to-Glycan Binding: Molecular Recognition through Polyvalent Interactions Mediates Specific Cell Adhesion

**DOI:** 10.3390/molecules26020397

**Published:** 2021-01-13

**Authors:** Gradimir Misevic, Emanuela Garbarino

**Affiliations:** 1Research and Development, Gimmune GmbH, Baarerstrasse 12, 6302 Zug, Switzerland; 2School of Basic Medical Sciences, Nanjing Medical University, 101 Longmian Avenue, Jiangning District, Nanjing 211166, China; ema14@njmu.edu.cn

**Keywords:** glycans, glycan-to-glycan binding, molecular recognition, polyvalent interactions, glyconectins, Le^x^, carbohydrates, carbohydrate-to-carbohydrate interactions, glycocalyx, cell adhesion and recognition, glycan structure and function

## Abstract

Glycan-to-glycan binding was shown by biochemical and biophysical measurements to mediate xenogeneic self-recognition and adhesion in sponges, stage-specific cell compaction in mice embryos, and in vitro tumor cell adhesion in mammals. This intermolecular recognition process is accepted as the new paradigm accompanying high-affinity and low valent protein-to-protein and protein-to-glycan binding in cellular interactions. Glycan structures in sponges have novel species-specific sequences. Their common features are the large size >100 kD, polyvalency >100 repeats of the specific self-binding oligosaccharide, the presence of fucose, and sulfated and/or pyruvylated hexoses. These structural and functional properties, different from glycosaminoglycans, inspired their classification under the glyconectin name. The molecular mechanism underlying homophilic glyconectin-to-glyconectin binding relies on highly polyvalent, strong, and structure-specific interactions of small oligosaccharide motifs, possessing ultra-weak self-binding strength and affinity. Glyconectin localization at the glycocalyx outermost cell surface layer suggests their role in the initial recognition and adhesion event during the complex and multistep process. In mammals, Le^x^-to-Le^x^ homophilic binding is structure-specific and has ultra-weak affinity. Cell adhesion is achieved through highly polyvalent interactions, enabled by clustering of small low valent structure in plasma membranes.

## 1. Glycan-to-Glycan Molecular Recognition Is Biologically Relevant Biopolymeric Interaction

Intermolecular recognition between various types of biopolymers is enabling the selective building of self-assembling structures and their hierarchical organizations into complex systems possessing catalytic properties. These processes are therefore considered essential for the emergence, evolution, and sustained existence of diversified life forms.

Establishing the causal relationship between a variety of biological functions of biopolymers with their structures, required the multidisciplinary approach combining chemistry, biology, physics, mathematics, and informatics, together with their associated technologies. Numerous resulting studies revealed that the specific type of a biological function is based on a particular structure present in a biopolymer, which is providing the necessary selectivity of intermolecular binding. This now obvious principle of structure to molecular recognition to biological function relationship applies to all types of interactions among biopolymer, such as nucleic acids, proteins, glycans, and lipids, as well as for biopolymers interactions with oligomers and with non-polymer molecules of either organic or inorganic nature.

The best documented biopolymeric interactions with molecular recognition properties related to a specific biological function are protein-to-protein, protein-to-glycan, protein-to-lipid, lipid-to-lipid, protein to nucleic acid, and nucleic acid to nucleic acid-binding. The knowledge that also glycan-to-glycan binding can be specific in nature and designated as a molecular recognition process with functional significance is slowly but progressively emerging [1]. From a chemical and biochemical viewpoint, it is noticeable that the nature of this type of intermolecular association, often neglected and least studied, must differ from those of other biopolymer classes. Physicochemical properties of glycan biopolymers, reliant on their monosaccharide building blocks and their extended linear and often branching sequences with repeating structural motifs, are fundamentally different from those of linear polymers of protein composed by amino acids, of nucleic acids build by nucleotides, and of lipids hydrocarbon chains [2,3,4]. These facts provide the chemical rationale that glycan-to-glycan molecular recognition is based on a different type of molecular mechanism enabled by particular chemical properties of glycan structures, their valency and the affinity of their functional binding sites. Therefore, biological functions based on glycan-to-glycan intermolecular associations should extend and complement the spectrum of well-known selective bindings among proteins, nucleic acids, and lipids.

Glycan-to-glycan binding is also referred to as carbohydrate-to-carbohydrate interactions, as well as self-associations of carbohydrates [1]. The biological relevance of this conceptually new intermolecular association with a high degree of specificity was shown to mediate cell adhesion and recognition in two experimental model systems, sponge species-specific cell aggregation, which is designated as evolutionary simplest self-non-self-recognition [1], and mice embryonal cell adhesion and metastasis [5]. Therefore, glycan-to-glycan binding is an addition to protein-to-protein and protein-to-glycan bindings. The last two types of intermolecular interactions were for a long time recognized as the only two existing mechanisms underlying the vast of physiologically relevant cellular interactions. Now all three binding types are considered to be essential in (a) evolution of multicellularity, founded on the emergence of the first self-recognition system that is envisaged as the primordial immune system, (b) acquired and innate immunity, enabling self-non-self-recognition, (c) morphogenesis during embryonal development, (d) recycling, regeneration and healing of tissues, (e) pathological conditions such as metastasis following malignant transformation, and (f) infectious diseases associated with invasion of viruses, bacteria, fungi, and multicellular parasites. Multistep cellular interactions, requiring sufficient binding strength between specific structures with strictly regulated spatiotemporal expression, are fundamental for the life-sustaining physiological processes.

Glycan chemistry and biochemistry, entailing glycan isolation, analytics, sequencing, modification, conformation determination, and modeling studies, are more complex in comparison to other biopolymers [4]. Experimental procedures are also considerably slower, particularly for the case of glycan sequencing, which is not fully automated. Thus, linking a particular structure to a certain molecular mechanism enabling a specific function requires more time for glycans than for proteins, nucleic acids, and lipids. Consequently, the available literature on the subject of cellular interactions mediated by specific glycan-to-glycan binding is not proliferative and abundant as for cellular interactions mediated by protein-to-protein and for protein-to-glycan binding. As indicated in the above paragraph, the unfortunate consequence is that the functional role of the glycan-to-glycan mediated cell recognition is fully proven only for two types of molecules, large glyconectin (GN) glycans in sponges [1] and small Le^x^ trisaccharide epitopes in mice [5]. Indeed, it remains to be experimentally shown how universal is this type of molecular recognition in a variety of existing physiological processes requiring cell adhesion and recognition. The increase in understanding and appreciation of glycan-to-glycan binding in cell interactions, beyond the two examples studied so far, can only be achieved by applying the methods and the knowledge of biology, physics, chemistry, and informatics.

The structural aspects underlying the concept of glycan-to-glycan based molecular recognition is still better understood and more appreciated by chemists rather than by biologists. Chemist adopted and used the idea that highly polyvalent interactions between glycan polymers reach high specificity and affinity by utilizing carbohydrate-binding units displaying ultra-week binding forces such as hydrogen bonding, ionic interactions, hydrophobic or hydrophilic forces, and van der Waals forces [6]. This resulted in building a large number of self-assembling supramolecular structures useful in nanotechnology. Functional aspects of glycans’ role as mediators of specific cellular interactions are obviously the biologists’ field. Consequently, several types of glycan-to-glycan interactions such as cellulose, glycogen, galactans, alginates, fucoidans, xylans, lactose, and various N-linked glycan structures were demonstrated. They provide important structural information about the glycan-to-glycan binding but were not shown to mediate cellular recognition and adhesion, which is the focus of this review.

Involvement of heterophilic GM3 to Gg3 glycan-to-glycan interactions in cell adhesion of mouse B16 melanoma and T-cell lymphoma, and B16 and SPE-1 endothelial cell are cell adhesion culture model systems that are only indirectly connected to the in vivo observed physiological cellular interactions and thus will not be reviewed here in detail.

The above introductory description of general principles of biomolecular associations in life processes and of the novel case of glycan-to-glycan association mediating cell recognition and adhesion will be extended with the following two subsections. The first one will cover in-depth explanations for the rationale of the “unusual” glycan-to-glycan intermolecular binding. The second subsection will provide experimental evidence that connects the structural properties of glyconectins and Le^x^ glycans with the mechanism of molecular recognition underlying their biologically relevant functional role in cellular interactions in sponges and mice. The second subsection will end with the summarized comparison of the molecular mechanism of glycan-to-glycan binding to well-known protein-to-protein and protein-to-glycan binding.

The available knowledge about the structure to function relationship of glycan-to-glycan binding reviewed here is intended for the scientist interested in glycobiology and cell recognition and adhesion fields. The aim is to provide complete consolidated information and to inspire further research on glycan-to-glycan molecular recognition in cellular interactions.

## 2. The Rationale for Glycan-to-Glycan Binding Concept in Cellular Interactions

Cellular interactions in multicellular organisms are complex and multistep events resulting in cell recognition and adhesion that are keeping anatomic integrity, distinguishing self from non-self, enabling reproduction and morphogenesis, sustaining dynamic physiology of life, and playing an important role in some pathological cases such as metastasis and fighting infections of pathogens. Extensive multidisciplinary studies of molecular mechanisms underlying these processes revealed two types of structure-specific intermolecular associations with functional relevance: (a) protein-to-protein binding and (b) protein-to-glycan binding. Biochemical and cell biology research dedicated to characterization and measurements of specificity of protein-to-protein binding identified cadherin, integrin, and immunoglobulin families of cell adhesion molecules [7,8]. These three families of molecules are mediating a variety of cellular interactions either through homophilic binding (intermolecular binding between two identical molecules located on different cells, e.g., cadherin-to-cadherin) or heterophilic binding (intermolecular binding between two different molecules located on different cell, or cell and extracellular matrix, e.g., integrin to fibronectin). Studies on the second type of protein-to-glycan molecular recognition identified several lectin families of cell adhesion molecules and their corresponding glycan ligands with specific sequence motifs. Both types of intermolecular recognition were shown to operate via single or low valent interactions with a moderate or high binding affinity of a single site [9,10].

The alternative third type of molecular recognition concept based on structure selective glycan-to-glycan binding operating in cell recognition and adhesion emerged in 1987 [1]. This first report was initially followed by a relatively low number of publications when compared with other glycobiology topics. This research was either ignored or approached by skepticism. The common and somewhat justified questions raised were: (a) “Why should glycan-to-glycan binding exist when we have protein-to-protein and protein-to-glycan molecular recognition?”, and (b) “What are the molecular mechanisms of glycan-to-glycan binding that can provide necessary specificity and affinity?” Fortunately, sustained research efforts from a few groups dealing with this third type of glycan-to-glycan molecular recognition in cellular interactions provided structure to function related evidence leading to a higher level of acceptance of this new concept.

In this section, the detailed answers to the above questions will be covered by providing the rationale and experimental evidence that have supported the initial idea of glycan-to-glycan molecular recognition as a functional basis for cellular interactions. The first one is based on the fact that glycans are highly abundant at the outmost layer of plasma membranes, thus being the first and unavoidable molecular encounters of the environment. The second one is the spatiotemporal control of the expression of extremely variable sequences of glycan polymers and their particular physicochemical properties. Thus, both can provide the structural basis for the novel molecular recognition mechanism operating through proposed highly polyvalent interactions with often unmeasurably low-affinity of a single binding site [1].

### 2.1. Topology, Abundancy and Spatiotemporal Control of Glycans Expression on Cellular Membranes is Relevant for Glycan-to-Glycan Binding Mediated Cellular Interactions

#### 2.1.1. Topology and the Abundancy of Glycans on Plasma Membranes

The first step in the process of establishing contact between cells approaching each other must involve their outermost molecular layer. This layer was shown to be mainly composed of glycans and was named glycocalyx (Figure 1). Electron and optical microscopy imaging of cells revealed that the glycocalyx layer has a thickness ranging from a few hundred nanometers to millimeters and contains large amounts of densely packed acidic glycans (Figure 1) [11,12,13,14]. However, this layer is only visible after using specific fixation protocols with cationic dyes, which preserve and stains acidic glycans. Standard fixation and visualization procedures commonly applied in cell adhesion and recognition imaging-based studies result in the complete loss of these molecules through washing. The obtained images of cells are missing this glycan layer and are leading to the conclusion of their nonexistence in spite of their high abundance. Similarly, biochemical procedures that are usually used to analyze cell adhesion molecular components of plasma membranes suffer from the inability to isolate and analyze extremely large glycosaminoglycan and glyconectin acidic glycan types with a molecular mass over 10 × 10^5^ D. Therefore, many imaging, biochemical, and cell biology approaches were missing these large acidic glycan molecules in their structural and functional cell adhesion and recognition assays.

Isolation and characterization of large cell surface-associated glycoconjugates with molecular masses over 10 million Daltons, such proteoglycan carrying glycosaminoglycans, proteoglycan-like glyconectins carrying large fucosylated acidic glycans, and mucins with small O-linked acidic glycans are different from those applied to smaller glycoproteins N- and O-linked glycans. The large glycoconjugate biopolymers were identified as the major molecular components of the glycocalyx. The extension of cell adhesion transmembrane proteins from the plasma membrane is ranging from 5 to 20 nm, whereas acidic glycans of proteoglycans, glyconectins, and mucins are extending from the membrane at least 200 nm (Figure 1).

All known cell surface-associated glycans, large glycosaminoglycans, their more structurally complex relatives glyconectins, and small glycans are covalently attached either to transmembrane or extracellular matrix proteins and/or to membrane lipids, with the exception of hyaluronic acid. Extracellular glycoconjugates are indirectly associated with other membrane proteins via protein-to-protein or protein-to-glycan interactions. Although, they may also be using glycan-to-glycan interaction to anchor to membrane glycolipid or to glycan moiety of transmembrane glycoproteins. It must be noted that besides densely packed glycan, the outmost layer also has protein cores of proteoglycans, proteoglycan like glyconectins, and mucins that may participate in intermolecular associations. However, numerous structural analyses (NMR, X-ray crystallography, electron and atomic force microscopy, enzymatic probing of accessibility of protein part by enzymes) and modeling studies showed that this protein part is sterically more difficult to access by other proteins or glycan molecules. This is due to the shielding effect of multiple copies of large acidic glycans per protein core and even more densely packed small acidic glycans per mucin protein core, which are dominating the molecular space of membrane proteins (Figure 1). Therefore, similarly, as in the case of plasma membrane perspective view, here from more detailed individual molecular view, again encountering the protein core of glycoconjugates must pass through dense shielding barrier of their glycan moieties. Since all cell has comparable membranes architecture in concern to gross molecular organization of glycocalyx, although not the same molecular composition which is important for recognition function, it is unlikely to avoid glycan-to-glycan interactions as the first molecular binding or repulsion events during cellular interactions.

Taking into account that several types of large and small acidic glycans are highly abundant and tightly packed at the outermost layer of the cellular membrane, the twisted question arises: how could other transmembrane cell adhesion molecules position closer to the lipid bilayer of the plasma membranes function? Even if there is no glycan-to-glycan interaction, somehow, this layer must open the space to allow protein-to-protein and protein-to-glycan contacts that are located at the proximal vicinity of the lipid bilayer. Here the possible answer is proposed: the initial step in cell interactions would occur through glycan-to-glycan binding, which brings the cell membrane closer and opens space for secondary protein-to-protein and protein-to-glycan interactions.

The fact that acidic glycans are expressed in large amounts at the outermost layer of cell surfaces and are thus the primary and unavoidable dense shield encountering environment validates the first rational of glycocalyx glycan-to-glycan binding in cellular interactions (Figure 1).

#### 2.1.2. Spatiotemporal Control of Glycan Expression

Glycan biosynthesis occurs intracellularly through a multistep of enzymatically catalyzed processes [4]. The exact molecular pathways differ in prokaryotes and eukaryotes but are universally based on glycosyltransferase activities at the specific cellular localizations generating primary glycan sequences [4,16]. Therefore, glycan structures encoding is principally different from that of protein-encoding mechanism, where the sequence is generated via transcription and translation of nucleic acids code using a specific codon for each amino acid. The conclusion is that structures of glycans are determined indirectly by the expression and spatial positioning of specific sets of glycosyltransferase enzymes.

Many of the studied glycan structures were shown to have cell and tissue-type specific expression, as well as species-specificity [17,18]. This research was based on either isolation of glycans directly or by testing for the presence of particular types of glycosyltransferases and their enzymatic activities. Furthermore, biosynthesis of glycan is also regulated in time, which is limiting their occurrence to the particular stage of embryonal development, physiological states such as hematopoiesis, immune response and inflammation, and in some pathological cases, particularly in tumor growth and metastasis [4,19].

Cellular recognition and adhesion require the presence of a specific set of cell adhesion molecules that display molecular recognition either as homophilic or heterophilic binding with spatiotemporal regulation of their expression. These have been shown for cell adhesion molecules operating through protein-to-protein binding and for lectins operating through protein-to-glycan binding [20,21]. Since the availability of specific glycan structures as described above is also often restricted to sets of specific cell types in the time-controlled fashion, the second rationale for glycan-to-glycan molecular recognition guiding specific cellular interaction is also fulfilled.

### 2.2. Diversity of Glycan Sequences Related to Specific Glycan-to-Glycan Binding in Cellular Interactions

Glycans are either linear or branched biopolymers of monosaccharides present in all living unicellular and multicellular organisms. Since the glycan classifications were extensively reviewed [4], it will only be summarized in order to facilitate the understating of semantics related to the structure and function relationship for glycans. Glycan classifications are commonly based either on their structural properties, although also functional properties like involvement in cell adhesion and recognition, being blood groups and antigenic markers for developmental stages and tumors are also used. The first structural category is based on the size and/or their degree of polymerization, which is separating them into polysaccharides (n > 10, large glycans) and oligosaccharides (n > 2 < 10, small glycans). The second one is the branching property dividing glycans into linear and branched types. The third one is distinguishing homopolymer (polysialic acid, glycogen, cellulose, etc.) from heteropolymer glycans. The fourth one is based on the presence of the ionic charge, which allows categorization into acidic (ionic) and neutral glycans and is often extending to subdivision based on the type of charge. The fifth one concerns the nature of chemical groups, which is substituting monosaccharides N-acetylation (N-acetylated glycans) and sulfation (sulfated glycans). The sixth structural category considers the conjugation linkage-type to proteins, such as N linkage to asparagine (N-linked glycans) and O linkage to serine and threonine (O-linked glycans), and linkage to lipids. A further subdivision is based on the type of protein and lipid conjugate.

Diversity of glycan structures, as well as of any other biopolymers such as protein and nucleic acid, depends on (a) a number of different monomer units that can be used to build polymers via enzyme catalysis in cells, (b) possible sequence and size variation of polymers, (c) a number of possible linkage types between two monomers, (d) presence of branching option in polymers, and (e) post biosynthesis modification. Before considering a more detailed evaluation of the structural diversity of biopolymers and their differences, it is important first to distinguish theoretically possible diversity from the experimentally found variability. Second, it is necessary to take into consideration that the level of the structural variability depends on cellular localization, physiological state, and developmental stage of the organism within the life cycle. Third, several proteins and glycan structures were conserved in evolution; however, some are species-specific and can play an important role in xenogeneic self-non-self-recognition [4,18]. Hence, total discovered diversity in all living organisms for a particular biopolymer is obviously larger than that found in a single species.

The number of different monomer units that can be metabolically used to build the biopolymers in a human cell is 4 nucleotides for nucleic acids, 20 amino acids for proteins, and a surprisingly large number of 776 monosaccharides entries are cataloged in data banks for glycans [22]. In *Homo sapiens,* the most commonly found monosaccharide building blocks for glycans are 6 hexoses, 6 hexosamines, 10 acidic monosaccharides, and 3 pentoses, making a total number of 25 monosaccharides. Therefore, even limited to humans, theoretically and experimentally obtained evidence confirm that monomer building blocks for glycans exceed other biopolymers.

Taking the sequence and the size of biopolymers as determining parameters of their variability, proteins and nucleic acids are limited by the number of the existing sequences of the available genes coding for protein structures in each species. According to curated data from the Genome Reference Consortium Human Build 38 patch release 13 (GRCh38.p13) information, the estimated number of human coding genes is between 19,000 to 20,000 [23,24]. Around 28,000 different proteins have been experimentally identified in humans [25]. It should be noted that the posttranslational modifications, such as glycosylation, are adding to the protein diversity. Glycan sequences and glycan sizes are not directly limited by the number of genes due to the indirect control of their biosynthesis via glycosyltransferases expression and enzyme activity. Therefore, almost unlimited variations in sequences and sizes can be theoretically achieved by temporal and spatial control of only a few glycosyltransferases. According to CAZy, *Homo sapiens* have 243 sequences in 47 families of glycosyltransferases [16,26,27]. Theoretically, this number of enzymes allows the generation of enormous structural variability for glycans in somatic cells. In GlyGen [28] and UniCarb-DB Reference Collection [29] database, 15,069 glycan structures, ranging up to 37 monosaccharides, were found in humans. Currently, 77,495 glycan structures were reported in GlyTouCan data based on glycan size ranging up from 2 to 83 monosaccharides [30]. These data do not include very large glycans and may contain several synthetic glycan structures. Since the sequencing of the majority of such very large glycosaminoglycans and glyconectin acidic glycans, comprising of up to 1000 monosaccharides is not complete, and since they already revealed the existence of highly variable structures in spite being built by only a few monosaccharides, it is expected that the number of glycan structures present in humans as well as in other species would exceed that of proteins for each species. Similarly, small N- and O-linked glycans in animals, as well as large peptidoglycans and cell wall glycoconjugates in bacteria, cell wall carbohydrates in plants, and glycans in algae and fungi, show a high degree of diversity. According to carbohydrate structure database merged from bacterial, archaeal, plant, and fungal part (http://csdb.glycoscience.ru/database/ last updated 31 August 2020) it is estimated that total of 24,669 glycan compounds from 12,521 organisms [31].

The number of possible glycosidic linkages between two monosaccharides is ranging from 3 to 5. Each of these covalent bonds can have two anomeric configurations α or β. Taken together, the maximal number of possible structures between two D-hexoses A and B is 128. Two amino acids linked by the peptide bond allows 2 possible sequences designated as A–B and B–A structures. The same is true for nucleic acids. For the chain of three different hexopyranoses without repeats, e.g., (A; B; C), up to 6144 different glycan structures can be formed, whereas only 6 tripeptides are possible for three different amino acids without repeats, e.g., (A; B; C) [32].

Branching in biopolymer is observed only in glycans. They can be multiple and of complex configurations. This is allowing higher diversity of structures even with the small number of building blocks and a small number of enzymes involved in polymer biosynthesis.

While we are far from having the complete experimentally determined number of existing biopolymer structures in living organisms, from available genomic, proteomic, and glycomic datasets, it can be concluded that glycans are also extremely heterogeneous. In order to better understand and evaluate autologous (from the same individual), allogeneic (from different individuals of the same species), and xenogeneic (from different species) glycan heterogeneity, it is necessary to perform more sequencing in humans and a variety of other species, and to compare them with proteomic and genome data.

Summarizing the above discussion, it can be concluded that the molecular mechanism of generating glycan structural diversity is conceptually different from the diversity generation for the proteins. The idea about theoretically possible structural variability as well as experimentally shown diversity of glycans sequences is appealing and promising in regards to their anticipated functions in cellular recognition and adhesion driven through specific glycan-to-glycan interactions.

## 3. Glycan-to-Glycan Molecular Recognition Mediates Cellular Interactions via Highly Polyvalent and Strong Binding Based on the Ultra-Weak Affinity of a Single Site

The concept of the specific glycan-to-glycan binding mediating cell recognition and adhesion in sponges, the simplest multicellular organisms, was first discovered and published in 1987 [1]. The species-specific reaggregation of dissociated sponge cells showed that the large polyvalent glyconectin type of glycans are essential for self-recognition [1,33,34,35,36,37]. In 1989 the second model system using mice indicated Le^x^-to-Le^x^ binding as a possible basis for cell adhesion [5]. In the following years, more sophisticated structure to function related studies at the molecular and cellular level by quantitative measurements of binding strength and affinity, and cell recognition and adhesion assays provided solid evidence for glycan-to-glycan molecular recognition in the sponge and mice model system organisms.

Due to the difference in the chemical nature between proteins and glycans, the molecular basis determining the specificity of binding is fundamentally different. Proteins display homophilic and heterophilic interaction among themselves via the single monomeric binding site with high-affinity or low valency and moderate affinity per binding site [9,10]. The same principle of mono and low valency with high-affinity is true for protein-to-glycan binding [9,10]. Contrary, glycan-to-glycan binding is based mainly on homophilic and highly polyvalent interactions of a single site, which display as a monomer very low, or even no measurable binding affinity [1]. These findings were demonstrated by multidisciplinary research using electron, optical and atomic force microscopy imaging and measurements of binding strength on a single molecular level and by kinetic measurements of binding affinity using surface plasmon resonance, calorimetric measurements, and theoretical calculations based on results obtained by biomolecular simulation programs [1,33,34,35,36,37]. Ex vivo and in vitro cell and glycan functionalized bead adhesion, and recognition provided the final complementing results [33,34,38]. In the following subsections, all experimental approaches used for sponge and mice model systems will be reviewed and discussed in terms of how this type of molecular recognition operates in cellular interactions.

### 3.1. Glyconectin Type of Glycan-to-Glycan Binding Mediates Cell Recognition and Adhesion

#### 3.1.1. Structure

Glyconectin family of glycoconjugates, previously named “aggregation factors”, are structurally defined large biopolymers so far found in sponges [1,33,34,35,36,37,38,39,40,41]. The initial physicochemical characterization of *Microciona prolifera* glyconectin, the first known member of this family, defined it as a large proteoglycan-like molecule due to the molecular mass >10 × 10^6^ D, and presence of at least 50% of acidic glycans containing significant amounts of fucoses, uronic acids, and sulfated and/or pyruvylated hexoses and hexosamines [1,33,34,35,36]. This unique monosaccharide composition is different from classical glycosaminoglycans, which contain an N-acetylated or N-sulfated hexosamine and either a uronic acid (GlcA or IdoA) or galactose [42]. Glyconectin glycans did not cross-react with anti-glycosaminoglycan antibodies and were not digestible by enzymatic treatment with specific glycosaminoglycan hydrolyzing enzymes [1,33,34,35,36,38]. Thus, the fundamental differences must exist in sequences between glyconectin glycans and common glycosaminoglycan families of large acidic glycans. Partial glycan sequencing of glyconectin glycans from *Microciona prolifera* (GN 1), *Halichondria panicea* (GN 2), *Cliona celata* (GN 3), using NMR and mass spectrometry in combination with chemical fragmentation fingerprinting, revealed species-specific sequences with common novel structural properties of repeating oligosaccharide motifs (Figure 2, Figure 3, Figure 4 and Figure 5) [33,34,35,36]. Species-specific aggregation was also confirmed with *Haliclona occulata* (GN 4) and *Mycale fusca* (GN 5) [1,43], in which sequencing and characterization remained unpublished. Following studies on *Suberites suberia* (GN 6), *Ficulina ficus* (GN 7), *Desmapsamma anchorata* (GN 8), and *Spongilla alba* (GN 9) glyconectins and their glycans revealed three new members of the glyconectin family [39,40,41,44]. Structural analyses and sequencing of glyconectin glycans are not yet complete due to their extremely large size and the lack of specific glycosylhydrolases, leaving chemical degradations as the only fragmentation possibility.

Electrophoresis and column chromatography purification of total glycan fraction from glyconectins isolated from GN 1, GN 2, and GN 3 showed that each GN is composed of species-specific sets of large acidic glycans (Figure 2) [33,34,46]. Common features of GNs in different sponge species are the multimillion molecular mass and a large number of glycan copies per glyconectin molecule. In GN 1, GN 2, and GN 3, about 20 copies of 10 to 20 × 10^5^ D acidic glycans were found [33,34]. GN 1 had also about 20 copies acidic g6 glycan of 6 × 10^3^ D [33,34]. Purification of all of the glycans allowed structural analyses, direct functional testing for glycan-to-glycan binding, and raising of specific monoclonal antibodies.

Scanning probe microscopy and electron microscopy imaging of sponge glyconectins GNs showed partial resemblance to proteoglycan shapes with up to 1 μm in length [37,39,47]. They are composed of multimeric core protein in elongated or circular form with about 20 long glycan chains with sizes up to 200 nm (Figure 3) [37]. These observations correlate with the quantitative biochemical measurements of the number of g200 glycan copies per glyconectin molecule.

Raising the battery of monoclonal antibodies against the most studied sponge glyconectin (GN 1) from *Microciona prolifera* lead to the discovery of Block 1 and Block 2 antibodies, both able to block species-specific adhesion mediated by GN 1 [1,43,46,48]. The Block 1 monoclonal antibody recognizes trisaccharide D-Gal*p*4,6 (R) Pyr β 1–4 D-Glc*p*NAc β 1–3 L-Fuc*p* [33,34,35,36,43,46,48] (Figure 4). The Block 2 antibody recognizes a disaccharide structure D-Glc*p*NAc3S β 1–3 L-Fuc*p* present in the large g200 glycan [33,34]. This was shown by immunoblotting of electrophoretically separated GN 1 glycans and isolated di- and trisaccharide fragments by their immunoprecipitation and ELISA assays [33,34,35,36,46] (Figure 4 and Figure 5). These results indirectly suggest the functional role of these glycan sequences in cell adhesion.

Quantitative biochemical analyses of the isolated di- and trisaccharide after GN 1 glycans chemical degradation and determination of the number of Block 1 and Block 2 antibodies binding site per single GN 1 molecule showed that both structures repeat up to 2000 times per GN 1 and 100 times per g200 glycan [3,34,43]. Similar to the revealed high polyvalency of GN 1 g200 glycan and its disaccharide, the quantitative analyses of GN 2 and GN 3 acidic glycans, named g180 (molecular mass 1.8 × 10^5^ D) and g110 (molecular mass 1.8 × 10^5^ D), and their fragments, showed about 20 repeats of glycans per GN and up to 100 repeats of sequenced motifs of oligosaccharides per glycan [33,34,43]. Therefore, it can be concluded that two levels of valency must be considered for the disaccharide and trisaccharide structures: a) the number of copies per glycan and b) the number of glycan copies per glyconectin.

#### 3.1.2. Function and Mechanism of Binding

Sponges were one of the first model system used to study species-specific recognition and adhesion [47,49,50,51,52,53,54,55]. They are the simplest multicellular organisms that are the closest living descendants of primordial multicellular life forms. For this reason, sponges can be considered as the ideal xenogeneic self-recognition experimental model organism. The over 100-year-old discovery of species-specific reaggregation of dissociated sponge cells inspired cell biologists and biochemists to search for the molecular basis of species-specific cellular interactions. Results were identification of glyconectins cell recognition and adhesion molecules forming the thick glycocalyx layer of sponge cells.

The advantage of the marine sponge reaggregation over other cell adhesion models is an unnecessary enzymatic treatment to dissociate cells. Just lowering the concentration of Ca^2+^ in seawater, combined with the mild mechanical squeezing of small pieces of sponges through 100 μm nylon mesh cloth, results in complete nonenzymatic dissociation. After reading Ca^2+^ back to the cells, reaggregation will occur in a matter of minutes, and in a few days, a functional adult sponge will be fully regenerated (Figure 6).

When dissociated cells from two or more species are mixed, they will reaggregate in a species-specific manner (Figure 6). The key experiments leading to the identification of glyconectins involved their isolation from the dissociated cells. This was achieved by washing away glycocalyx glyconectins from the cell surface in Ca^2+^ free seawater and collecting them in supernatants. The next step was Ca^2+^ precipitation of glyconectins from supernatants, followed by further purifications using ultracentrifugation, column chromatography, and/or electrophoresis.

Reaggregation experiments used cells from different sponge species depleted from their endogenous glyconectins. In order to stop metabolic reconstitution of glyconectins on the cell surface, cells were either fixed or kept at +4 °C. When Ca^2+^ was added back to such glyconectin depleted cells, aggregation did not occur. Only by reconstituting seawater with either supernatants collected after cell washing, which are containing glyconectins, or by adding purified glyconectins from each species would initiate sponge reaggregation in a species-specific manner (Figure 6) [33,34,38,43,47,49,50,51,52,53,54,55]. Gelation assays using isolated glyconectins showed species-specific self-interaction only in the presence of a physiological concentration of calcium ions indicating homophilic binding. These experiments provided direct evidence of Ca^2+^ dependent glyconectins to glyconectin binding in cell recognition and adhesion. Until 1980 glyconectins were only purified but were neither sequenced nor completely analyzed. This justified at that time their name as undefined “aggregation factors”. The new glyconectin name (connection of cells via glycans) defines more precisely this class of molecules on the basis of their structure and function.

Detailed structure to function related experiments were following the initial reaggregation studies. The aim was to dissect the exact molecular mechanism of glyconectin mediated cell recognition in sponges. The approach, as described above, was to isolate and sequence GN glycan and to prepare anti-GN glycans monoclonal antibodies in order to, directly and indirectly, test their function. The essential adhesion experiments were performed by coating color-coded beads with either glyconectins from different species through linking protein part to the beads and leaving glycans free to interact, or by coating the same type of beads with isolated glyconectin glycans. The species-specific bead aggregation for both types of beads, GNs and GN glycans, was initiated only in the presence of Ca^2+^ (Figure 7) [1,33,34,38,43]. The beads also co-aggregated in a species-specific manner with dissociated sponge cells (Figure 8).

The Block 1 and Block 2 monoclonal antibodies that recognize highly repetitive D-Gal*p*4,6 (R) Pyr β 1–4 D-Glc*p*NAc β 1–3 L-Fuc*p* and D-Glc*p*NAc3S β 1–3 L-Fuc*p* structures, respectively, present in GN 1 g200 glycan, selectively blocked *Microciona prolifera* cell reaggregation, GN 1 and GN 1 glycan coated beads adhesion (Figure 6, Figure 7 and Figure 8) [1,33,43]. These results confirmed that homophilic and highly polyvalent glycan-to-glycan binding in all sponge species tested is the basis for species-specific cell recognition and adhesion (Figure 6, Figure 7 and Figure 8). Furthermore, these results directly prove that specific glycan sequences are mediating self-recognition. Since all of the tested GN glycans have highly repetitive oligosaccharide structures, and since the exact functional disaccharide sequence of GN 1 was identified to mediate cell recognition and adhesion via self-interaction, it can be concluded that glycan-to-glycan molecular recognition is based on the highly polyvalent (up to 100-valent) interactions.

In spite of partial sequencing of four different glyconectin glycans, direct evidence for the structure to function relationship was provided only for two disaccharide sequence D-Gal*p*4,6 (R) Pyr β 1–4 D-Glc*p*NAc β 1–3 L-Fuc*p* and D-Glc*p*NAc3S β 1–3 L-Fuc*p* of GN 1 g200 glycan. GN glycans from other sponge species show glycan-to-glycan self-recognition by using different sequences, but the functional sequences remain to be completed.

Although cell and bead experiments directly revealed the molecular recognition mechanism of glycan-to-glycan binding in cellular interactions associated with self-recognition in sponges, it was necessary to obtain additional quantitative data of their self-associations. For that purpose, atomic force microscopy (AFM) measurements of the binding strength between individual GN molecules under physiological solution were performed [37]. The experimental procedure of covalently crosslinking of a single GN molecule via protein part to the cantilever tip and to surfaces of mica, as an oriented monolayer, required deposition of chromium and gold, followed by the formation of a self-assembled monolayer of 11,11-dithio-bis-(undecanoic acid N-hydroxysuccinimide ester). The formed succinimide groups on surfaces were used for crosslinking of the protein part of GNs. This procedure insured shielding of ionic interaction between inorganic surfaces, as well as covalent crosslinking of protein parts in a properly oriented fashion. As a result, glycan chains remained freely exposed on surfaces [37]. Therefore, the obtained AFM measurement results can be interpreted as the binding force between single pair of GNs via their glycan-to-glycan interactions and not between other inorganic interactions of surfaces or detachments and reattachments of molecules, which can occur if not covalent crosslinking is used.

AFM measurements under physiological conditions showed that two GN 1 molecules bind to each other via their glycans with a force of up to 400 piconewtons [37]. This force was observed only in the presence of physiological Ca^2+^ concentration. As in the cell and GN beads aggregation, other bivalent cations could not replace calcium ions. Furthermore, cell adhesion inhibitory Block 2 monoclonal antibody, recognizing D-Glc*p*NAc3S β 1–3 L-Fuc*p*, which is enabling glycan-to-glycan interactions, completely blocked GN 1-to-GN 1 binding in AFM measurements [37]. These results indicated the direct role of sulfated disaccharides in GN 1 self-interactions. The obtained AFM results showed that a single pair of GNs could hold the weight of 1600 cells (Figure 9). It can be concluded that glycan-to-glycan intermolecular binding forces indeed represent the basis for the integrity and self-recognition of the multicellular sponge organism.

AFM measurements of binding strength between individual GN molecules enabled more detailed analyses of unbinding between one pair of glycan molecules. As shown in Figure 9, GN 1 displays multiple discreet steps of unbinding at the level of around 20 piconewtons of the applied force, and sometimes also at the level of over 50 piconewtons. The smaller steps can be interpreted as the unbinding of individual pairs of the D-Glc*p*NAc3S β 1–3 L-Fuc*p* sequences. The larger steps may involve larger blocks containing multiple binding sites, also including D-Gal*p*4,6 (R) Pyr β 1–4 D-Glc*p*NAc β 1–3 L-Fuc*p* sequences. Such discrete unbinding events present in AFM force-distance curves occur up to 200 nm, which is the length corresponding to the size of the single g200 glycan. The above-described AFM experiments at the single molecular level confirm biochemical measurements and prove specific and high binding strength based on the highly polyvalent self-binding of glycan structures, with weak binding strength of the single site.

Few groups followed the initial studies, also using AFM measurements for GNs and GN glycans [40,41,56]. Although a similar approach and comparable results were obtained for glycan-to-glycan specificity and binding force, the required procedures of functionalizing surface for covalent attachment of GNs and GNs glycans were not always followed. Therefore, caution of interpreting some of the reported data must be taken into account since simple adsorption was used instead of covalent crosslinking and since shielding of inorganic surfaces with lipid self-assembling monolayers was omitted. These may have led to the production of inconclusive results [40,41,56]. Furthermore, most of the studies using smaller glycan molecules result in the accommodation of several glycans molecules per tip because AFM tip size was larger than the size of a single molecule. Hence, AFM measures multiple molecular interactions, not the single ones [40,41,56]. This was not the case for GN 1 because AFM tips used can accommodate only one GN 1 molecule by crosslinking it solely via protein part.

The following important approach for investigating glycan-to-glycan self-association was done with synthetically prepared sulfated disaccharide, pyruvylated trisaccharide, and other control saccharides as multimeric neoglycoconjugates of BSA as carrier [57]. The synthetically obtained sulfated disaccharide is recognized as the natively prepared fragment of GN 1 g200 glycan by the adhesion inhibitory Block 2 monoclonal antibody. Self-binding kinetics of multimeric neoglycoconjugate (D-Glc*p*NAc3S β 1–3 L-Fuc*p*) was measured by surface plasmon resonance. The results showed that self-binding specifically requires the presence of 10 mM CaCl_2_ and that, as it could be expected, it is simultaneously self-aggregating in solution and binding to the functionalized sensor. Therefore, the affinity could only be estimated (ka = 10^2^ M^−1^, and an affinity, or avidity, of Ka = 10^5^ M^−1^) which is about 50 times higher than that of the binding between single Le^x^ epitopes (Ka = 2–3 M^−1^). Since all measurements are done in seawater with high ionic strength (0.5 M NaCl) and since sulfated and carboxylated conjugates used as a control did not show similar behavior, it could be concluded that self-interactions are structure-specific and not just simple ionic attraction [57]. The molecular mechanism proposed was that the stable octagonal or hexagonal coordination of the calcium ion occurs through three or four interactions from each disaccharide epitope. The AFM measurements with the same synthetic self-binding disaccharide showed calcium ion-dependent self-interactions for the sulfated disaccharide and not for self-interaction of the pyruvylated trisaccharide. In addition, heterotypic binding between di- and trisaccharide was not detectible [58]. Unfortunately, experiments were done in non-physiological low ionic strength (water and only CaCl_2_) and thus cannot provide a complete and conclusive interpretation of glycan-to-glycan binding under natural physiological conditions [58].

Colloidal force microscopy was used to measure the dynamic strength of the individual self-binding disaccharides of the GN 1 g200 glycans [59]. Sulfated disaccharide and non-sulfated control disaccharides were coupled to membrane-coated surfaces in order to mimic native cell-to-cell contacts. The binding strength and calculated affinity were measured as a function of calcium ions and loading rate. Obtained data were analyzed using a deterministic model for estimation of the basic energy landscape and the number of bonds involved in binding [59]. The modeling results indicated the equilibrium off-rate of k_off_ ≈0.0015 s^−1^ and a potential width of x^u^ = 0.25 nm. The same modeling approach was used to interpret binding data under different loading rates of applied rupture force. It could be estimated that the binding force per disaccharide linked to the lipid membrane in the physiological solution would be around 30 piconewtons. Although these results are in agreement with the first AFM measurements of GN 1 self-binding [37], physiological significance cannot be implicated due to the undisclosed buffer composition, which may not mimic physiological conditions.

The second D-Gal*p*4,6 (R) Pyr β 1–4 D-Glc*p*NAc β 1–3 L-Fuc*p* sequence was also prepared synthetically. Gold linked trisaccharide self-interactions was evaluated in water and 10 mM CaCl_2_ by NMR spectroscopy. These data were associated with Molecular Dynamic modeling, which was also done in water. This simulation predicted at the atomic level possible binding sites of calcium ions to monosaccharide components of the tested synthetic self-interacting trisaccharide [60]. Suggested was parallel orientation between two trisaccharides during the binding event (Figure 4). Both NMR and simulation were neither using the physiological ionic concentration of 0.5 M NaCl nor proper buffering, which are both essential natural conditions for calcium ions dependent glycan-to-glycan interactions.

Although several of the above-reviewed results of binding force and kinetics measurements between synthetic oligosaccharides are not mimicking the natural ionic conditions, they are very important since they reveal the possible structural details of glycan-to-glycan sequence-specific molecular recognition at the atomic level. Whether the same glycan configuration occurs during species-specific cell aggregation in the marine sponge *Microciona prolifera* via highly polyvalent and Ca^2+^ dependent self-binding of sulfated disaccharide D-Glc*p*NAc3S β 1–3 L-Fuc*p* and self-binding of pyruvylated trisaccharide D-Gal*p*4,6 (R) Pyr β 1–4 D-Glc*p*NAc β 1–3 L-Fuc*p*, still must be investigated using experimental physiological condition, as in the initial AFM experiments on GN 1 [34].

Highly polyvalent, structure-specific, and strong glycan-to-glycan binding, based on ultra-low-affinity (K*d* > 10 mM M) for a single homophilic binding site, can be envisaged as the Velcro-like interaction between the cell surfaces. The physiologically important consequence is the robust but reversible adhesion enabling a high degree of cell motility, which are resembling the Velcro-like peeling. The other two types of intermolecular interactions, protein-to-protein and protein-to-glycan, used by the common cell adhesion molecules, are based on strong (K*d* < 100 nM) or moderate (K*d* 100 nM to 10 μM) affinity of a single or few binding sites [9,10]. The result is low reversibility and necessity for recycling or degradation of adhesion molecules during cell migration.

In spite of the progress in proving the concept of glycan-to-glycan recognition mediating species-specific cell adhesion in sponges, future research needs to deliver the three-dimensional structure of self-interactions for sulfated disaccharides of g200 GN 1 glycan, as well as for other GNs. Complete sequencing of GNs and clarification whether two interacting polyvalent acidic glycan chains bind to each other in parallel or antiparallel (opposite directionality) fashion is also required. If g200 glycans on the same GN 1 molecule bind in a parallel orientation, then all binding sites will be used intramolecularly. However, even in the parallel mode, the high polyvalent long glycan chains could create complex and multiple networks of inter and intramolecular glycan interactions.

One of the proposed working models for GN 1 glyconectin mediating cell recognition and adhesion can be envisaged as two events (Figure 10). The first one is structure-specific g200 to g200 homophilic and highly polyvalent self-binding of about 100 sulfated disaccharides and about 100 pyruvylated trisaccharides. The exact position of disaccharide and trisaccharide in g200 glycan is not yet revealed. The second step in cell adhesion is the binding of GN 1 to the plasma membrane via g6 glycan using highly polyvalent and specific interactions with either a transmembrane lectin protein or glycans of transmembrane proteins (Figure 10) [53,54,60]. Although the exact mechanism for other GNs must be established in more detail, GN mediated species-specific cellular interactions were directly shown to be based on highly polyvalent glycan-to-glycan interactions (Figure 10).

The fact that GN glycans are the dominant component of the sponge cells glycocalyx and are binding to each other with the strong force inspired the hypothesis that the cell recognition and adhesion are involving multiple types of molecular binding events in an orchestrating way. GN type of glycan-to-glycan interactions may decrease the distance between adhered cells and concentrate cell surface glycans into a smaller space. These would permit the secondary step of cell adhesion mediated by binding involving glycan to protein and protein-to-protein interactions. Indeed, several types of lectin and cell adhesion molecules have been actually identified in sponges; however, only a few lectins in some sponge species were experimentally shown to play a role in cell recognition [61].

Glyconectin type of glycan-to-glycan molecular recognition through homophilic highly polyvalent interactions was only investigated and shown in sponges. Although some preliminary biochemical and immunological data have revealed that glyconectin type of glycans is present in mammals [48] and in some lover invertebrates [62], this promising and appealing concept must be experimentally further investigated in order to prove to be universally used in other multicellular organisms.

Schematic presentation of the glycocalyx layer in Figure 10 is drawn to scale. It is based on the microscopic and X-ray crystallographic data of plasma membrane biopolymers. Since glyconectin type of glycans, together with mucins and proteoglycans, are the most abundant and highly extended glycoconjugates from the plasma membrane (>200 nm), it should be stressed that this type of intermolecular binding is envisaged just as the first step of cellular interactions which involves a variety of cell adhesion molecules operating via protein-to-protein and protein-to-glycan interactions based on microscopical and X-ray structural analyses. These three types of intermolecular recognition processes are coordinated and should be taken as complementary to each other. It could be hypothesized that these three types of binding are occurring during all physiologically relevant cellular interactions, even in the immune system dealing with the fine-tuning of self and non-self-recognition. Finally, glycan-to-glycan mediated recognition in sponges, the simplest multicellular organism, indicates that this interaction may also be important for the evolution of more complex living organisms than sponges.

### 3.2. Le^x^ Type of Glycans-to-Glycan Binding Mediates Cell Recognition and Adhesion

#### 3.2.1. Structure

Lewis x (Le^x^) or CD 15 determinant is the terminal trisaccharide glycan structure with sequence D-Gal*p*(β 1–4)[L-Fuc*p*(α 1–3)]GlcNAc(β 1-)-R [63] (Figure 11). It was described in 1978 as part of the stage-specific embryonic antigen (SSEA-1) using the monoclonal antibody approach [64,65]. The carbohydrate determinant was found in glycolipids and milk oligosaccharides in *Homo sapiens* and *Mus musculus*. According to the structural glycan database of GlyGen organization [66], the Le^x^ motif was found to be present in 820 molecules ranging in size from 37 to 4 monosaccharides in *Homo sapiens*. It should be taken into account that many of these structures may not have Le^x^ at the nonreducing terminal.

Le^x^ is expressed in a stage-specific manner on 56 different glycoproteins, including mucins and 764 different types of glycolipids [66]. Both types of glycoconjugates are localized in plasma membranes of different cell types such as cancer cell, embryonic cell, epithelial cell, leukocyte, granulocyte, monocyte, dendritic, neurons, and stem cells [66]. In total, 1825 structures contain the Le^x^ motif [67].

#### 3.2.2. Function and Mechanism of Binding

After experiments done in 1978 by Solter and Knowles, by raising specific monoclonal antibody recognizing SSEA-1, several reports emerged showing the involvement of this, now named Le^x^ trisaccharide, in cellular interactions related to the compaction process of the mouse embryos at the morula stage [64]. Le^x^ was recognized as one of the strongly regulated stage-specific embryonal structures in mice [65]. It was later shown that it plays an important role also in tumor and neural cell adhesion. Initially, the exact molecular mechanism was not revealed. One of the postulated and recognized Le^x^ binding receptors was CD2 (glycan to protein) involved in T and NK cellular interactions [68]. In 1989, following the report on glyconectin type of glycan-to-glycan binding in sponge cell recognition system [1], Le^x^-to-Le^x^ interactions were published, providing experimental evidence that this homophilic glycan-to-glycan binding could mediate tumor cell adhesion [5,69]. Studies were based on in vitro model system of Ca^2+^ dependent aggregation of mouse teratocarcinoma F9 cells. First, Le^x^ was shown to be expressed on F9 cell glycoproteins by cross-reactivity with the anti-Le^x^ antibody. Second, aggregation could be specifically inhibited only by lacto-N-fucopentaose III, which has Le^x^ epitope. Third, liposomes containing Le^x^ displayed self-aggregation in the presence of calcium and magnesium ions. Fourthly, Le^x^ liposomes and F9 cells showed specific binding to Le^x^ coated plastic surfaces. These results suggested that Le^x^-to-Le^x^ binding may play an important role in cell recognition of F9 cell aggregation as well as during embryonic development in mice.

An additional set of important experiments showing the specific glycan-to-glycan interactions of Le^x^ were also performed with mouse embryonic stem cells. The compacted cells stage occurs only when they are expressing Le^x^, and decompaction will occur only in the presence of trivalent Le^x^ and not in the presence of other trivalent Lewis type oligosaccharides [5,69]. These results revealed that Le^x^ terminal trisaccharide is a functional adhesion structure operating via self-interaction. However, the molecular nature of its carrier was unknown. The continued search identified polylactosaminoglycan with a high mass range rather than expected glycosphingolipid in F9 cells. This molecule named “embryoglycan” demonstrated self-aggregation only in the presence of Ca^2+^, as GN in sponges. Defucosylation resulted in the loss of autoaggregation, indicating the role of fucose in Le^x^-to-Le^x^ binding. It is interesting to notice that GN and Le^x^ have similarities in terms of fucose being the necessary function required monosaccharide. However, also marked differences exist in terms of the presence of charge in GNs and no charge in Le^x^, as well as the high valency in GNs, mainly appearing in long extended form, versus small size and lower valency in branched form of Le^x^.

Two cell line model systems, F9, and embryonal stem cells express Le^x^ and cell adhesion molecule E-cadherin. Both molecules are also expressed during embryogenesis and required calcium ions for promoting cell adhesion. Knocking out the E-cadherin gene by homologous recombination in both F9 and ES cell lines did not result in the loss of cell adhesion, indicating the involvement of Le^x^ [70].

The specificity of Ca^2+^ mediated homotypic interaction between two Le^x^ was approached by preparing lipid vesicles functionalized with glycolipids bearing monomeric or dimeric Le^X^ structure [71,72]. This method was selected because it mimics the natural environment of Le^x^ in the glycolipid from on plasma membranes. Using the micropipette aspiration technique with contact angle measurements, homotypic interaction was found to be weak but specific in the presence of calcium ions. In order to obtain the free energy of adhesion for Le^x^, data were modeled by taking into account various contributions of all vesicle glycolipid intermolecular bindings. This enabled to separated and quantified contribution of specific and nonspecific interactions. The specific adhesion energy for Le^X^-to-Le^X^ was calculated to be about 11 µJ/m^2^ in 110 mM CaCl_2_ and 4 µJ/m^2^ in 200 mM NaCl_._, for lactose-to-lactose was 6 µJ/m^2^ in 110 mM CaCl_2_ and 9 µJ/m^2^ in 200 mM NaCl, and for lactose to Le^x^ was 2 µJ/m^2^ in 110 mM CaCl_2_ and 5 µJ/m^2^ in 200 mM NaCl. These results provided definitive evidence for CaCl_2_ dependent Le^x^-to-Le^x^ binding specificity and involvement in cell adhesion. Interestingly, a repulsive interaction was observed for dimeric Le^X^, indicating possible intramolecular Le^x^ association on the same glycolipid via calcium ions, resulting in no free Le^x^ for intermolecular binding.

Le^x^-to-Le^x^ interactions were simulated with molecular dynamics in order to obtain information about the geometry, the dimerization mechanism, and stoichiometry of interacting calcium ions [73]. Simulations were performed in the explicit solvent with and without calcium ions. Results showed that calcium favors the Le^x^ dimerization and occurs with a change of the free energy from −5.2 kcal/mol to −7.2 kcal/mol. The major energetical difference can be assigned to the solute electrostatic energy of −2.5 kcal/mol^−1^ with calcium and −0.5 kcal/mol without calcium. This was in agreement with experimental data. Hydrogen bonds were found to be identical with or without calcium, as well as the hydrophobic contribution to the solvation free energy. Finally, the performed simulation revealed two possible conformations of Le^x^ dimer interactions with calcium ions involving electrostatic forces.

Another type of molecular modeling using the Amber suite of biomolecular simulation programs was developed to simulate homotypic Le^x^-to-Le^x^ interactions in the explicit solvent with and without calcium ions [74]. The simulated water box had about 3000 water molecules, ten calcium ions, and 20 chloride ions resulting in a neutral global charge, and Le^x^ with initial conformational packing obtained from the crystallographic coordinates. The carbohydrate force field accepted theoretically for glycans was used for all simulations. The simulation results show that calcium ions are favorable for Le^x^ dimerization. A decrease in free energy was observed from −4.0 kcal/ mol to −10.0 kcal/mol, and in two different simulations set up decrease was from −1.7 kcal/mol to −8.3 kcal/mol. Like in the above-described modeling study, the electrostatic contribution seems to be important, as expected for cations. The solute entropic was not found to be significant. This theoretical study indicated that Le^x^ is valuable for exploring molecular behavior with calcium ions. However, like many other modeling approaches, it is limited by computation power and knowledge of the force fields to be used for mimicking natural ionic conditions.

The selective self-interaction of Ca^2+^ dependent Le^x^-to-Le^x^ was also measured by isothermal titration calorimetry in an aqueous solution of gold glyconanoparticles functionalized with Le^x^ [75]. Comparison of Le^x^ to lactose and maltose disaccharides showed that the process of binding is slow and takes place with a decrease in enthalpy of about 160 kcal mol^−1^. Lactose and maltose glyconanoparticles showed very low heat evolvement and quickly reached thermal equilibrium. AFM and electron microscopy monitoring of gold bead aggregation revealed a slow adhesion process. Measurements with Mg^2+^ and Na^+^ cations confirmed selectivity for Ca^2+^ indicating ion and structure-specific Le^x^-to-Le^x^ interactions. Since many measurements with glycan functionalized gold nanoparticles were apparently done in aqueous solution within 10 mM range of divalent cations, they only provide results valid for non-physiological conditions and leave an open question about interactions occurring in the natural environment. Therefore, new approaches are needed for measurements of oligosaccharides mediating glycan-to-glycan binding.

An integrated nanotechnological strategy with AFM, electron microscopy, and surface plasmon resonance, were used to quantify at the molecular level the binding force and affinity of Le^x^-to-Le^x^ interactions. This model system of gold glyconanoparticles and coating surfaces with lipid-linked oligosaccharides enable measurements with monovalent and polyvalent clustered Le^x^ mimicking the natural glycolipid membranes [76,77,78]. Such an approach is very important from the physicochemical point of view. However, the inherited problems of the gold nanoparticles’ stability, which is greatly dependent on the nanoparticle size, the size and charge of the polymer coating, and ionic strength of the buffer, may not always allow measurements condition comparable to the natural ones. Therefore, caution must be taken when translating results to the physiological conditions.

Le^x^, lactose, various control oligosaccharides, and lipids functionalized self-assembled monolayers of on surfaces and gold nanoparticles were used for plasmon resonance measurements of binding affinity between Le^x^. This model system was mimicking polyvalent glycosphingolipid membrane clusters. A high-affinity with the slow association and a gradual dissociation was observed. For the binding of multivalent Le^x^ nanoparticles to Le^x^ monolayers in the presence of calcium ions, K_d_ of 5.4 × 10^−7^ M was calculated. The binding is of Le^x^ monomer to Le^x^ monolayers in the presence of calcium ion showed a very fast association and dissociation indicating a very weak interaction with K_d_ of 5.7 × 10^−3^ M. The affinity of polyvalent lactose interaction with Le^x^ monolayer showed K_d_ of 80 × 10^−3^ M in the presence of calcium ions. Reverse experiments using polyvalent lactose gold nanoparticles to surface-attached Le^x^ showed similar K_d_ values at the range of 10 × 10^−3^ M. Since no binding was observed in the absence of calcium ions, and since these data showed that high polyvalency of ultraweak monovalent self-interactions of Le^x^, it could be concluded that with calculated − ∆G of 8.5 [kcal/mol], Le^x^-to-Le^x^ binding can indeed mediate cellular interactions. Binding of lactose glyconanoparticles to lactose monolayers showed a rapid association in the presence of calcium ions and a fast dissociation with the K_d_ of 1.5 × 10^−5^ M [76,77,78].

Experiments using the same gold glyconanoparticles as in surface plasmon resonance experiments showed self-recognition and aggregation in calcium-containing aqueous solution upon examination with transmission electron microscopy.

Atomic force microscopy measurements of Le^x^-to-Le^x^ binding strength were performed in water with and without calcium ions. The same type of lipid-linked Le^x^, lactose and other control structures of neoglycoconjugates were used as for the surface plasmon resonance experiment [77,78,79]. Le^x^ neoglycoconjugates was covalently attached via lipid linker to the AFM tip and to the flat surfaces. This procedure results in the creation of a densely packed Le^x^-lipid self-assembled monolayer. Since the size of the tip is considerably larger than a single molecule, AFM measurements between the functionalized tip and the functionalized surface will measure multivalent and not the single Le^x^ binding force. Statistical analyses of force-distance curves from hundreds of AFM measurements performed in the presence and absence of calcium ions in water both showed multiple unbinding events with the rupture force for the single event of about 20 piconewtons in the presence and absence of calcium ions. The multiple events of 20 piconewtons can be interpreted as polyvalent interactions. Control experiments using Le^x^ functionalized tips binding to lactose neoglycoconjugates linked to the surface did not show binding in the presence or absence of calcium ions. The same was found with lactose-to-lactose interactions. These results indicated that under low, non-physiological, ionic strength Le^x^-to-Le^x^ binding forces were, surprisingly, not calcium ion-dependent.

Later studies were using membrane probe force microscopy involving lipid bilayer vesicles functionalize with Le^x^-to-Le^x^ bilayer on the surfaces [80]. This is a more natural architecture of imitating plasma membranes than that reported in the previous studies [77,78]. The binding force of Le^x^-to-Le^x^ interactions was measured under more physiological conditions using ionic strength comparable to the physiological one. Force of about 120 piconewtons was obtained between for Le^x^-to-Le^x^ functionalized membrane-like bilayers in buffer containing calcium ions. The binding strength was 27 piconewtons in the absence of calcium ions. The control nonfunctionalized membranes unbinding force was about 40 piconewtons in the buffer with calcium ions. These experiments showed the calcium ions dependent Le^x^-to-Le^x^ binding. The binding force must have originated via polyvalent Le^x^-to-Le^x^ interactions since the contact zone of vesicles is large. Therefore, the exact binding force between individual Le^x^ molecules remains to be experimentally determined. The extrapolated and modeled data indicated ultraweak binding between individual pairs of Le^x^ molecules. Due to the lack of instrumentation sensitivity when applying physiological ionic strength and temperatures, measurements of single molecular interactions for Le^x^ may not be yet achievable.

Modeling studies based on AFM analyses of Le^x^-to-Le^x^ binding with using a least-squares fit of the binding energies obtained from the model to the experimental data, AFM binding force data, resulted in reaction rates of *k*_on_ = 18/second and *k*_off_ = 7/second. The binding energy of Le^x^-to-Le^x^ was estimated to be 1 *k*_B_*T* [80].

Experimental procedures used in AFM measurements of intermolecular binding strength have the inherent inability to mimic the complex natural conditions. This was often resulting in the failure to attribute the binding force to the molecule intended to be studied. Therefore, the observed similarities and/or differences in results obtained by different groups for glyconectins and for Le^x^ homophilic binding forces should be interpreted with caution. It is also important to take into account that AFM measurements of binding between molecules attached to any surfaces are collectively accessing the numerous interatomic interactions which are difficult to analytically resolve in time and space.

Le^x^ is localized on the plasma membrane as a terminal trisaccharide of various glycolipids and glycoproteins. Besides being involved in cell adhesion via homophilic glycan-to-glycan binding, Le^x^ also interacts with selectins during various types of cellular interactions. As for glyconectin homophilic interactions, the importance was to resolve Le^x^ parallel versus antiparallel self-binding.

Since glycosphingolipids are parallelly oriented in the outer layer of cell membranes, the cis-homophilic interaction between Le^x^ glycosphingolipids could stabilize Le^x^ microdomains. Antiparallel or trans-homophilic contacts would mediate the cell-to-cell adhesion [81,82]. NMR and modeling studies indicated that various possible conformation of Le^x^ with calcium ions could allow both cis and trans homophilic binding. This dual-mode interaction can stabilize cell-to-cell adhesion. Since this process is happening at very close proximity to the plasma membrane, Le^x^ glycan-to-glycan interactions could be the last step in cell adhesion, occurring after the initial glyconectin-to-glyconectin type of glycan interactions at the periphery of the glycocalyx, which is followed by cell adhesion molecules using protein-to-protein and protein-to-glycan binding (Figure 10).

## 4. Challenges

For the biologically oriented research, the challenging goal is to establish the structure to function relationship at the molecular and cellular level. To achieve this objective, it is necessary to acquire complete sequence and conformation of studied molecules and perform in vivo and in vitro quantitative measurements of their physiologically relevant functional properties. Following are the major challenges for the glycobiology field and particularly for the topic of glycan-to-glycan binding as the mediator of cell recognition and adhesion:a.Improvement of isolation technology for large acidic glycans;b.Automated and rapid high throughput sequencing of large acidic glycans;c.Synthesis of small and larger (polymeric) glycan sequences;d.Conformational studies;e.Quantitative measurements of intermolecular interactions under physiological conditions on the single molecular level;f.Labeling glycans and following their spatiotemporal distribution in living cells;g.Complete glycome analyses in healthy and pathological states at least in model system organisms;h.Spatiotemporal studies on the molecular and cellular level for the elucidation of the multistep nature of cell recognition and adhesion processes mediated by glycan-to-glycan, protein-to-protein and protein-to-glycan binding.

Resolving the above challenges in the field of glycobiology will result in a complete understanding of the biological role of glycan-to-glycan binding in cellular interactions.

## 5. Conclusions

The biological role of structure-specific glycan-to-glycan binding in cell recognition and adhesion was established for the large acidic glyconectin type of glycans in sponge xenogeneic self-recognition and for small neutral Le^x^ glycans in mice embryonal development and in tumor cell adhesion.

Partial sequencing, biochemical, immunological, and biophysical characterization of glyconectin glycans from several sponge species designates them to a novel class of large polyvalent (>100 repeats), polymeric (<100 kD), acidic, and fucose containing trisaccharide and disaccharide self-binding sequences. Glyconectin glycans are indirectly linked to the protein-bound to the plasma membrane.

Le^x^, contrary to glyconectin glycans, is a neutral and fucose containing terminal trisaccharide motifs on smaller glycans linked to lipid or transmembrane proteins.

Glyconectin glycans function as xenogeneic self-recognition molecules in sponges. The molecular mechanism is based on structure-specific, highly polyvalent, homophilic, and calcium ions dependent glycan-to-glycan binding. The homophilic binding strength between a single pair of large glyconectin glycan polymers measured with AFM under physiological conditions is very high, about 400 piconewtons, indicating that a single pair of glyconectins can hold the weight of 1600 cells. These measurements were the first direct evidence that cell adhesion can support functional integrity and self-recognition in multicellular organisms. Although the self-binding strength for the functional disaccharide and trisaccharide was not directly measured under physiological conditions, it could be deduced from the polymeric glycan self-binding to be below 30 piconewtons. The self-binding affinity of glyconectin functional disaccharide and trisaccharide sequences measured with surface plasmon resonance is low, 10^2^ M^−1^ and when extrapolated to polymers, it is, as expected, high, >10^5^ M^−1^.

The affinity between single Le^x^ epitope self-binding was ultra-weak, 2–3 M^−1^, (about 50 times lower than the self-binding between single pairs of glyconectin functional site). The exact binding force between a single pair of Le^x^ molecules remains to be experimentally determined by AFM. The extrapolated and modeled data from AFM measurements between multiple pairs of Le^x^ indicated ultraweak binding for the individual pair of Le^x^ molecules, probably below 20 piconewtons. The high valency of Le^x^ arises through their clustering in the membrane resulting in the high self-binding strength between clusters.

Structure-specific glycan-to-glycan binding in cellular interactions can be considered as a new emerging molecular mechanism complementing well-known protein-to-protein and protein-to-glycan binding. This molecular mechanism is based on highly polyvalent, high-affinity, and calcium-ion-dependent self-interactions of small glycan structures with the very low-affinity. Therefore, it conceptually different from that of low valency and higher affinity binding of protein functional binding sites mediating protein-to-protein and protein-to-glycan interactions.

Large glyconectin glycans’ high abundance and particular localization at the glycocalyx outermost cell surface layer indicates their functional role in the initial steps of cellular interactions followed by those of proteins localized closer to the plasma membrane. Le^x,^ in the form of glycolipids, is confined at the closest proximity of the plasma membrane and may be involved in the latest steps of cellular recognition and adhesion.

## Figures and Tables

**Figure 1 molecules-26-00397-f001:**
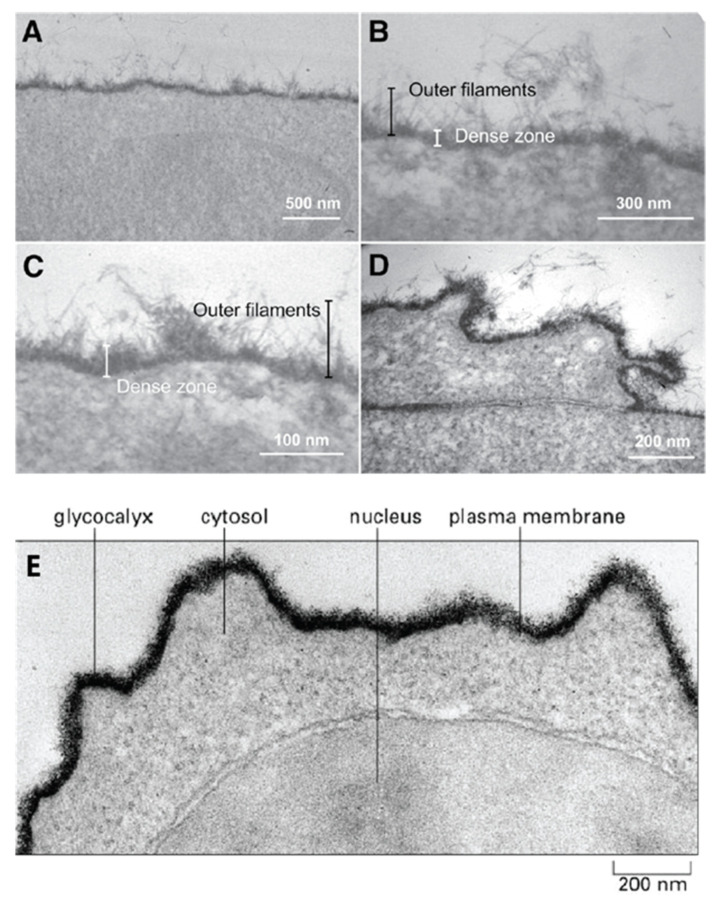
The endothelial glycocalyx on cultured human umbilical vein endothelial cells in vitro. (**A**) is an overview, (**B**,**C**) are closeups. Confluent overlaying cells are shown in (**D**) [15], human lymphocyte glycocalyx (**E**) from Molecular Biology of Cell, 4th Edition.

**Figure 2 molecules-26-00397-f002:**
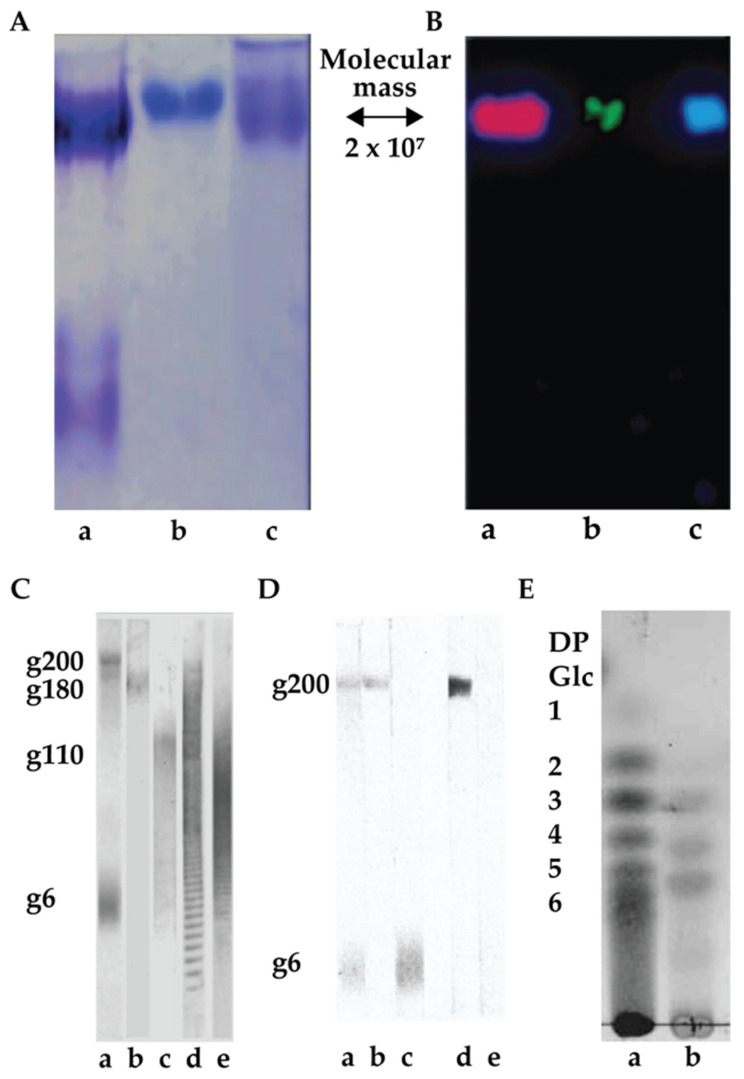
Biochemical analyses of sponge glyconectin GN 1–3 and GN 1–3 glycans. (**A**) Electrophoretic separation of sponge glyconectins on 0.75% agarose gel stained with 0.02% toluidine blue, followed by 0.1% Amido black 10B. Lane a, GNs from *M. prolifera* GN 1, lane b, from *H. panicea* GN 2, and lane c from *C. celata* GN 3 (10 μg each). (**B**) 0.75% agarose gel stained with color-coded fluorescent beads coated with lane a, GN 1 (pink), lane b, GN 2 (green), and lane c GN 3 (blue) in the presence of seawater (contains 10 mM CaCl_2_). Molecular weight was determined by ultracentrifugation. (**C**) Polyacrylamide gel electrophoresis of purified glyconectin glycan fraction. Electrophoresis of glyconectin glycans was performed on a polyacrylamide gradient gel (7.5–15%). Gels were stained with 0.3% Alcian blue in 3% acetic acid in aqueous 25% isopropanol. Lane a, 20 μg of GN 1 glycans; lane b, 20 μg of GN 2 glycans; lane c, 20 μg of GN 3 glycans; lane d, 50 μg of hyaluronic acid (Sigma, from bovine vitreous humor), 200 kDa, partially degraded; lane e, 50 μg of chondroitin sulfate (Sigma, from shark cartilage), 80 kDa [34]. (**D**) Separation of GN 1 glycans by electrophoresis on a polyacrylamide gel and immunoblotting of g200 by Block 2. 5 μL of GN 1 glycan samples were applied to a linear 7.5–20% polyacrylamide gel (molecular weight standards as in (**C**). After electrophoresis, gels were either stained with Alcian blue (a–c) or electroblotted to DEAE-nitrocellulose paper and decorated with antibodies (d and e). a, 10 μg of the total GN 1 glycans; b, 5 μg of the g200 glycan; c, 10 μg of the g6 glycan; d, 10 μg of the total GN 1 glycans blotted onto a DEAE paper and decorated with 2 μg of Block 2 antibody; and e, 10 μg of the total GN 1 glycans decorated with rabbit anti-mouse peroxidase-conjugated antibody, as control [43]. (**E**) TLC analysis of; standard glucose Glc degrees of polymerization (DP), and b, trifluoroacetic acid hydrolyzed fractions of GN 1 stained by sulfuric orcinol [33,35,36].

**Figure 3 molecules-26-00397-f003:**
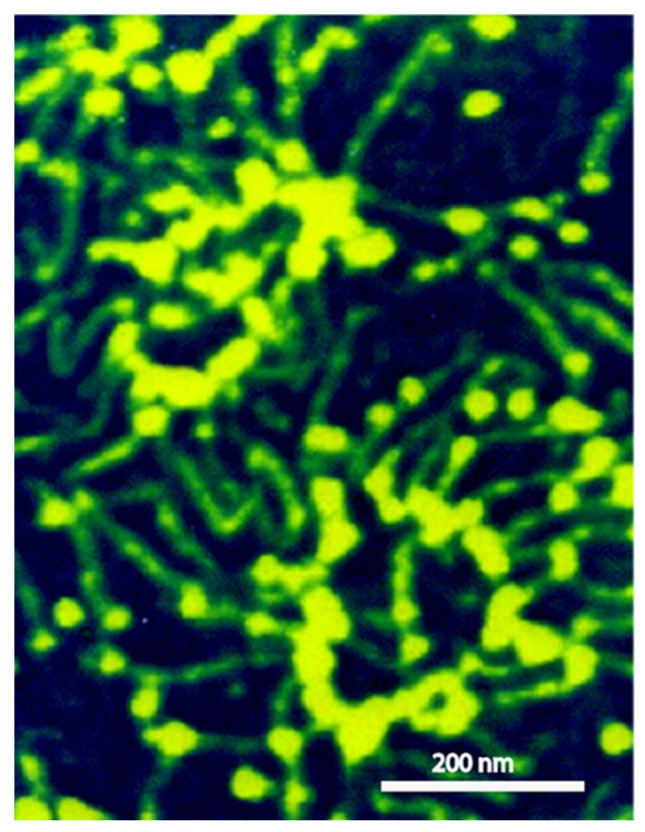
Atomic force microscopy image of glyconectin 1. The GN 1 (16 μg/mL) in seawater containing 2 mM CaCl_2_ were physisorbed to freshly cleaved mica (15 min), briefly rinsed with Nanopure water, and dried in air. Atomic force microscopy images (tapping mode) were taken in the air (see [37]).

**Figure 4 molecules-26-00397-f004:**
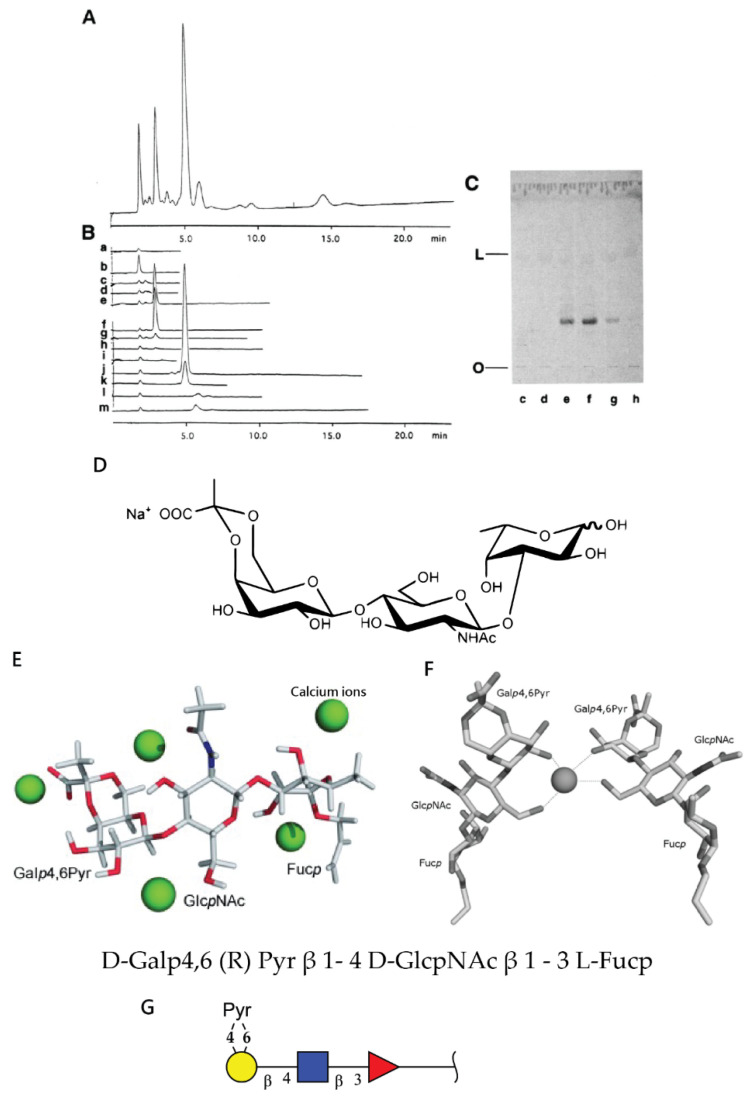
Purification of GN 1 derived oligosaccharides by HPLC. Pooled fractions of small oligosaccharides obtained after acid hydrolysis were separated on a semipreparative PA1 column, and fractions collected after online desalting. (**A**) An aliquot of pooled oligosaccharides run on a PA1 analytical column eluted isocratically with 80 mM sodium acetate, 10 mM NaOH, detector sensitivity 300 nA. (**B**) Aliquots of individual fractions (a–m), separated on the semipreparative PA1 column, rechromatographed using the analytical column under the same conditions as in (**A**). (**C**) Analysis of the oligosaccharides from (**B**) linked to dipalmitoyl phosphatidylethanolamine by TLC and immunodetection with monoclonal antibody Block 1. The positions of the original labeled as the letter *O* and free lipid labeled as the letter *L* are indicated on the left. (***D**,**E***) structure of GN 1 self-binding pyruvylated trisaccharide, (**F**) minimized average 3D structure of the second putative model for establishing pyruvylated trisaccharide-Ca^2+^-pyruvylated trisaccharide interactions as derived from solvated MD simulation. One calcium ion bridges the two pyruvylated self-binding trisaccharides through O2-Gal and O6-GlcNAc atoms at each sugar moiety [45], (**G**) symbolic representation of GN 1 self-binding pyruvylated self-binding trisaccharides [35,36].

**Figure 5 molecules-26-00397-f005:**
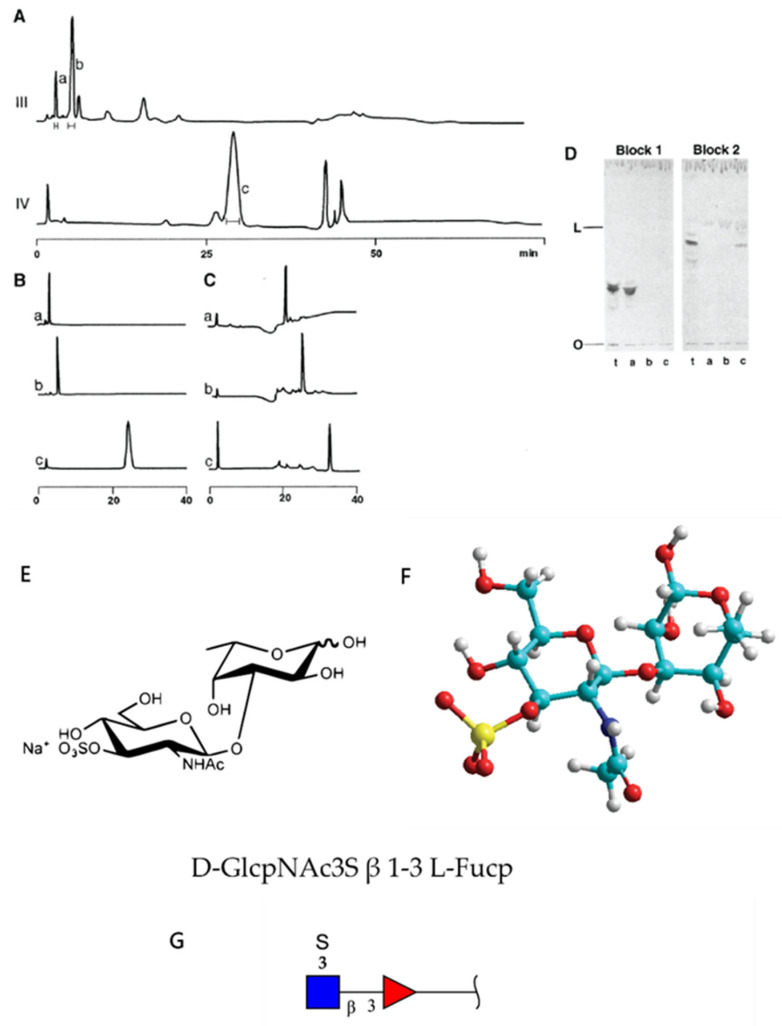
Purification of GN 1 derived oligosaccharides by HPLC. Pools III and IV of small oligosaccharides obtained after acid hydrolysis were separated on a semipreparative PA1 column, and fractions were collected after online desalting. (**A**) An aliquot of oligosaccharide pool III and IV run on a PA1 analytical column eluted isocratically with 80 mM sodium acetate, 10 mM NaOH. Peak fractions pooled are indicated by a *bar* and the letters a, oligosaccharide Block 1, b, oligosaccharide C-1, and c, oligosaccharide Block 2. (**B**) Aliquots of three individual oligosaccharides rechromatographed on PA1 analytical column, eluted isocratically with 80 mM sodium acetate, 10 mM NaOH, detector sensitivity 300 nA. (**C**) The oligosaccharides as in (**B**) analyzed with isocratic 100 mM NaOH and a gradient of 0–250 mM sodium acetate started at 10 min and completed at 30 min. (**D**) Analyses of total unfractionated oligosaccharides (*t)* and the purified oligosaccharides as neoglycolipids by TLC and immunodetection with monoclonal antibodies Block 1 and Block 2. The positions of the original labeled as the letter *O* and free lipid labeled as the letter *L* are indicated on the left. (***E***,***F***) structure of GN 1 self-binding sulfated disaccharide and (**G**) symbolic representation of GN 1 self-binding sulfated disaccharide [35,36].

**Figure 6 molecules-26-00397-f006:**
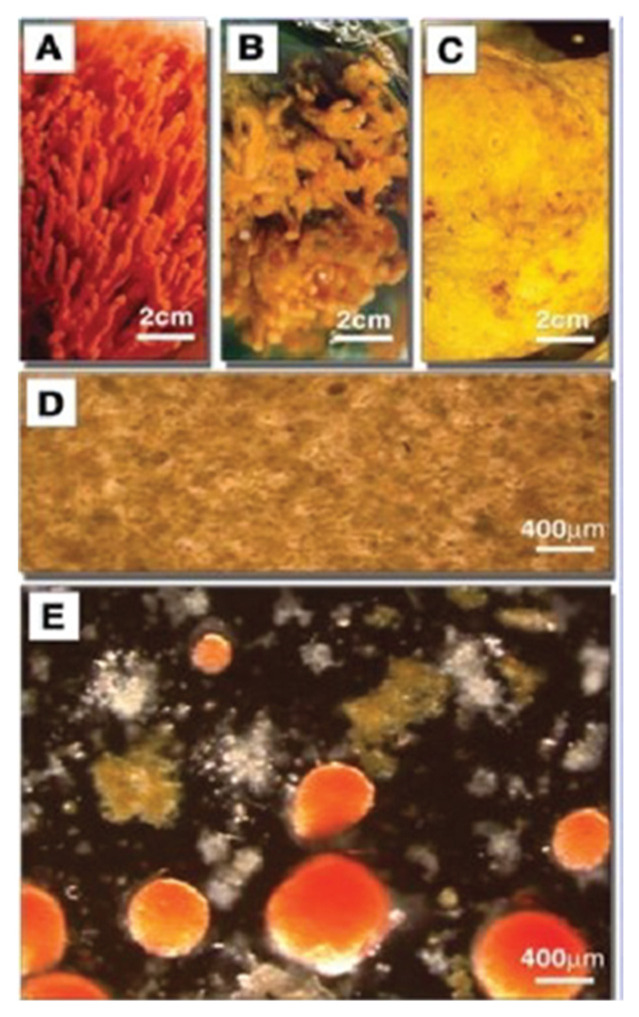
Glyconectin glycoconjugates are cell adhesion and recognition molecules. Ca^2+^-dependent glyconectin-to-glyconectin interactions mediate species-specific cell–cell recognition and adhesion. (**A**–**C**), *M. prolifera* (**A**), *H. panicea* (**B**), and *C. celata* (**C**) living sponges. Shown are self-and non-self-discrimination and adhesion in the suspension of mixed *M. prolifera* (orange), *H. panicea* (white), and *C. celata* (brown) live cells bearing glyconectins. (**D**,**E**), seawater without 10 mM Ca^2+^ (**D**) and seawater with 10 mM Ca^2+^ at 0 °C after 20 min of rotation (**E**). The microscopically observed color of the cells is somewhat different from that of the whole sponge. Early cell sorting experiments were usually done with binary sponge combinations at room temperature without rotation. The sorting is thus dependent on the presence of recognition molecules at the cell surface, cell motility, and speed of new synthesis and/or secretion of additional recognition molecules. Rotation assays using either metabolically attenuated or fixed cells reduce the number of variable parameters [34].

**Figure 7 molecules-26-00397-f007:**
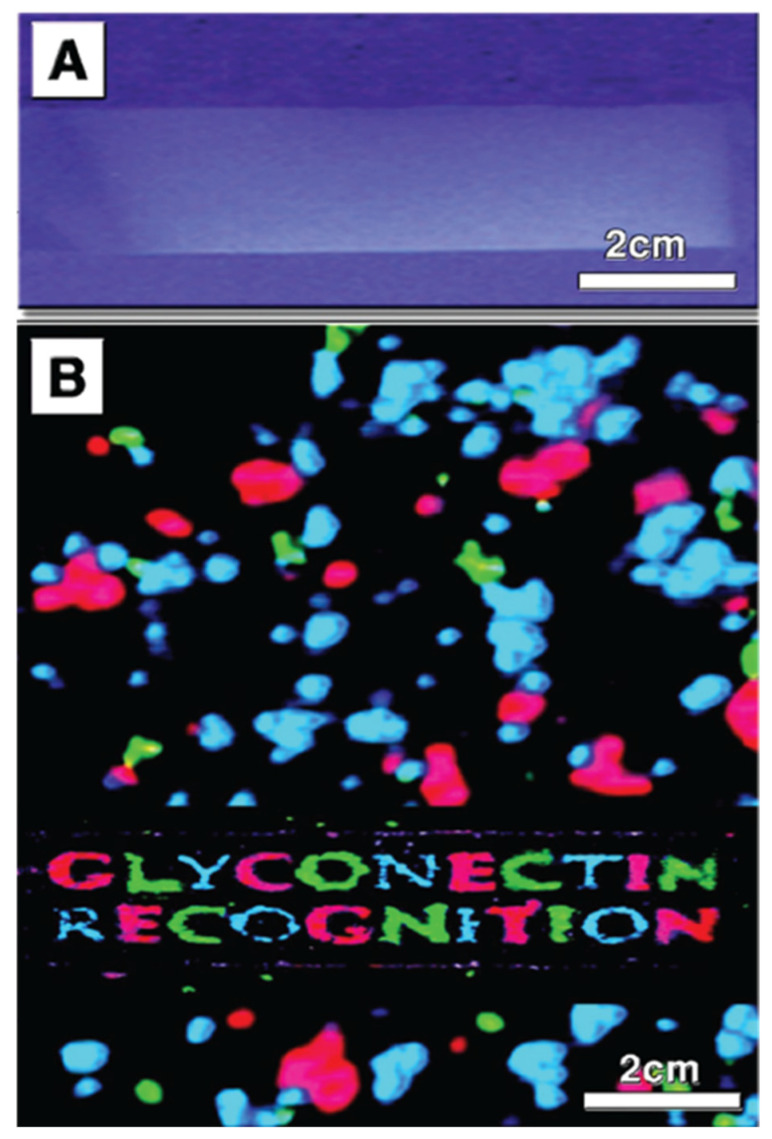
Simultaneous species-specific glyconectin-to-glyconectin recognition in suspension and blotting assay. Letters were drawn using 4 μL of 1.5 mg/mL glyconectins on a Hybond-C extra nitrocellulose membrane (Amersham Biosciences) and probed in seawater with pink, green, and blue fluorescent beads coated with glyconectin 1, 2, and 3, respectively. (**A**) seawater without 10 mM Ca^2+^. (**B**) seawater with 10 mM Ca^2+^. All photographs were taken after 30 min of mixing [34].

**Figure 8 molecules-26-00397-f008:**
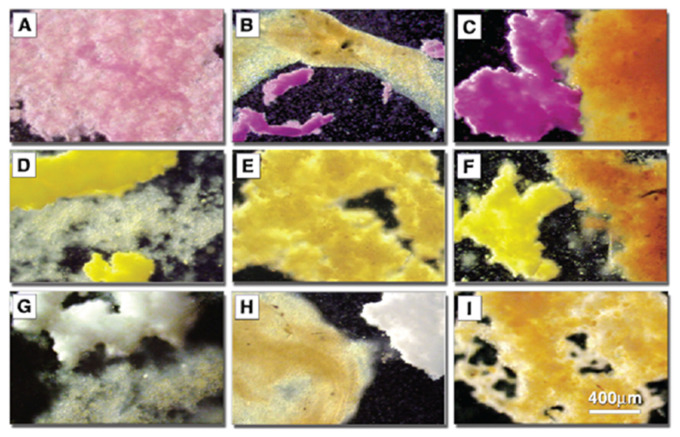
Species-specific glyconectin-to-glyconectin interactions mediate bead-cell recognition and adhesion xenogeneic glyconectin self-recognition in a mixture of glutaraldehyde-fixed cells and glyconectin-coated beads in to seawater buffered with 20 mM Tris pH 7.4 in the presence of 10 mM Ca^2+^. *M. prolifera* cells bearing glyconectin 1 were incubated with: glyconectin 1 (pink beads) (**A**), glyconectin 2 (yellow beads) (**D**), and glyconectin 3 (white beads) (**G**). *H. panicea* cells bearing glyconectin 2 were incubated with: glyconectin 1 (**B**), glyconectin 2 (**E**), and glyconectin 3 (**H**) color-coded beads. *C. celata* cells bearing glyconectin 3 were incubated with: glyconectin 1 (**C**), glyconectin 2 (**F**), and glyconectin 3 (**I**) color-coded beads (glutaraldehyde fixation changes cell colors, i.e., *M. prolifera*, orange to yellowish-white; *H. panicea*, white to yellowish-brown; and *C. celata*, brown to brownish orange. We did not observe differences in adhesion properties between fixed and live metabolically attenuated cells in a rotary assay [34].

**Figure 9 molecules-26-00397-f009:**
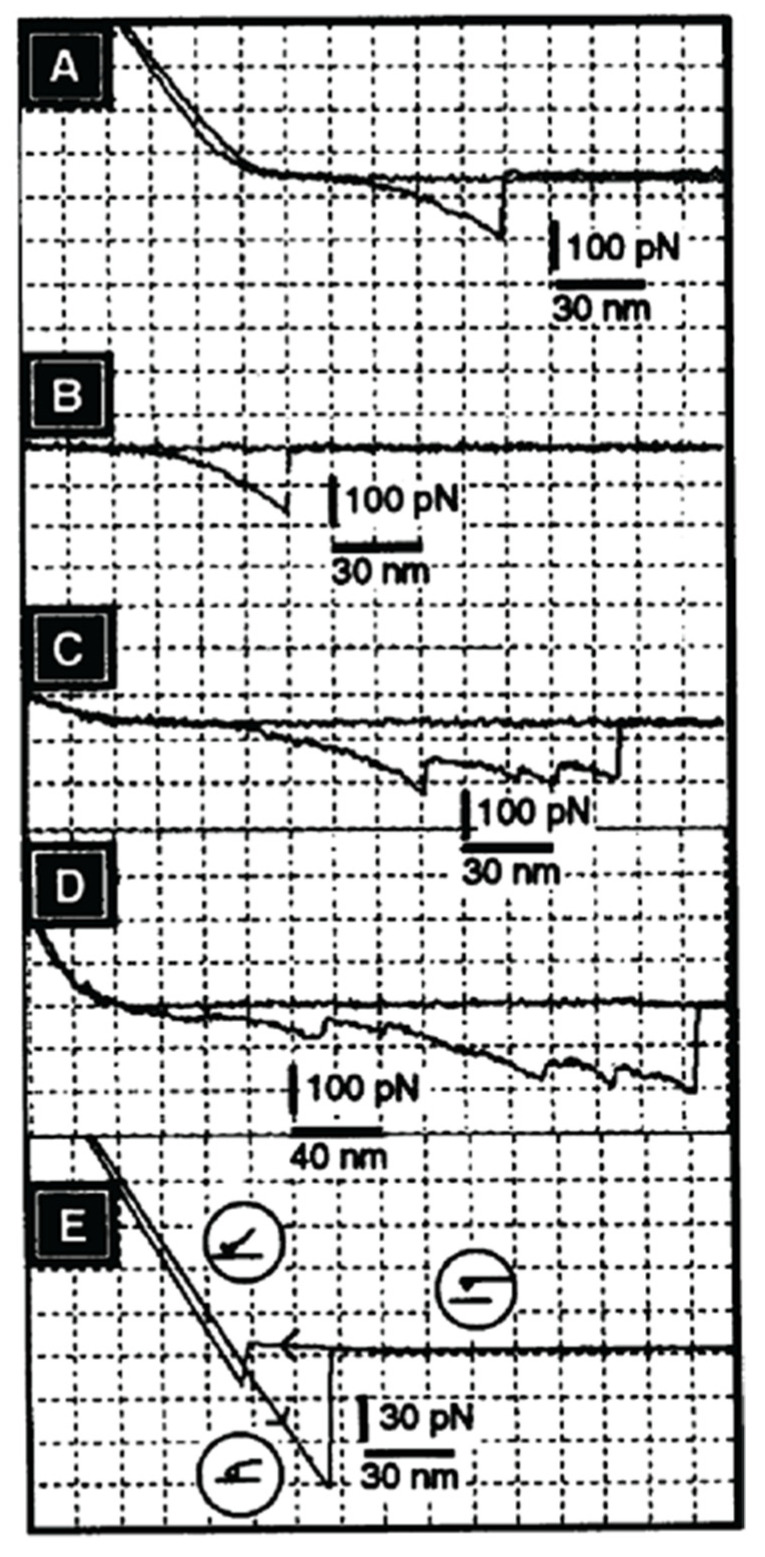
Typical atomic force microscopy (AFM) approach-and-retract cycles for GN 1-to-GN 1 interactions. The ***x***-axis shows the vertical movement of the cantilever; the *y*-axis shows the bending of the cantilever and thus the force acting on it. (**A**–**D**) represent typical GN 1-to-GN 1 binding, whereas (**E**) is an example of the interaction between two gold surfaces covered with self-assembled monolayers (1-dodecanethiol) [37].

**Figure 10 molecules-26-00397-f010:**
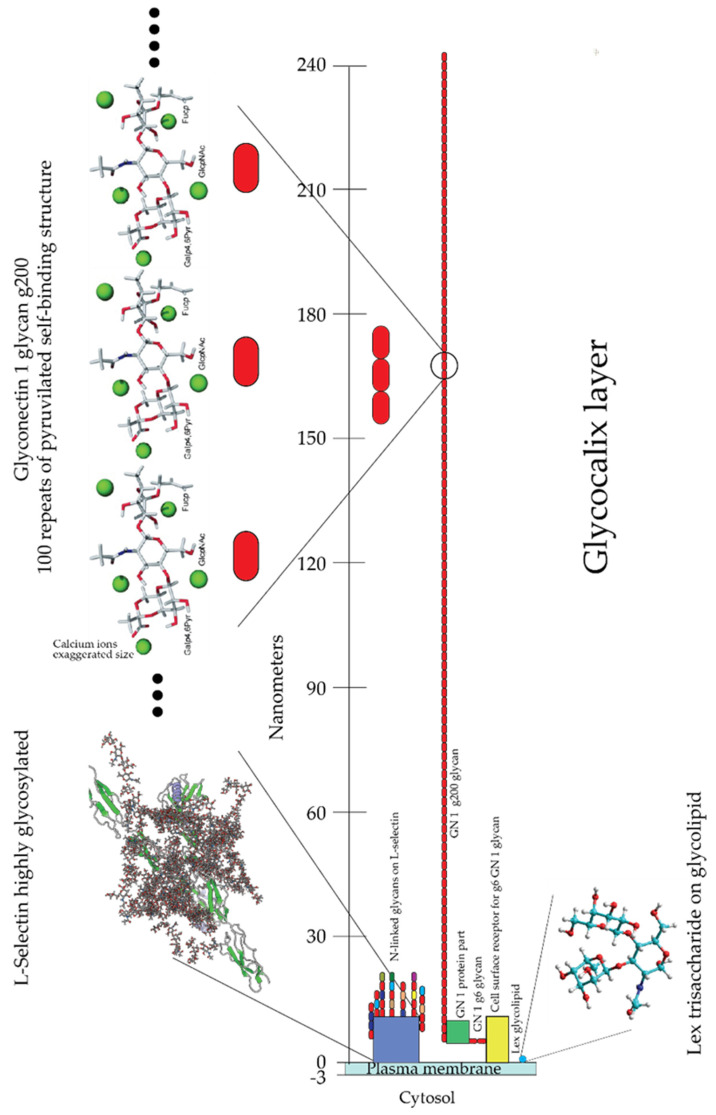
Schematic presentation of one part of the plasma membrane glycocalyx layer, containing one simplified structural representation of g200 glycan and one Le^x^, drawn to scale (exception is somewhat exaggerated calcium ion shell on GN 1 pyruvylated trisaccharide). 3D structures of pyruvylated GN 1 glycan and Le^x^ are obtained from modeling and NMR studies [60]. Glycosylated L-selectin is a hybrid structure from protein data bank combined with N-linked glycan structures builder GlyProt—in silico glycosylation of proteins.

**Figure 11 molecules-26-00397-f011:**
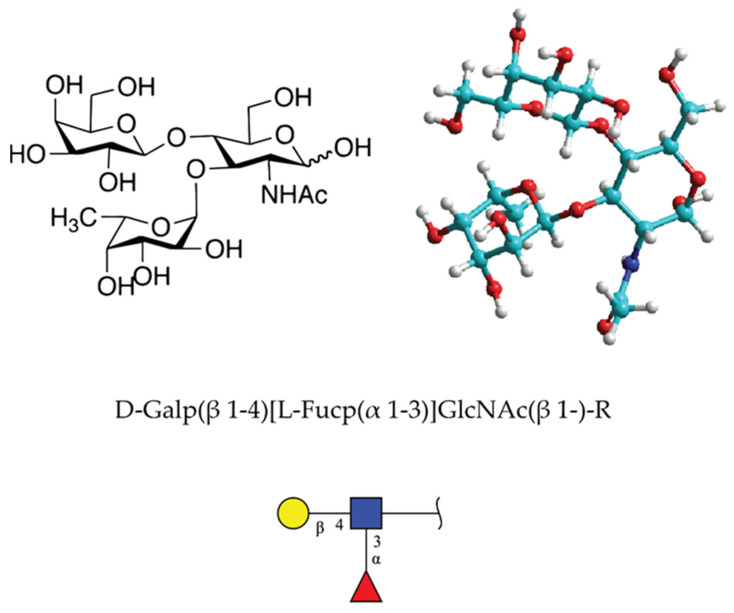
Le^x^ self-binding trisaccharide structure (**A**,**B**), and (**C**) symbolic representation of Le^x^ self-binding trisaccharide.

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
