# Peer review of "Glycan-to-Glycan Binding: Molecular Recognition through Polyvalent Interactions Mediates Specific Cell Adhesion"

_molecules, 2021, doi:10.3390/molecules26020397_

Round 1

Reviewer 1 Report

Comparisons between physiological and non-physiological conditions are forced and overwrought. Either the results are interesting but the nature of the conditions should be considered when interpreting or physiological conditions are necessary for the results to be valid. You can't say that the conditons are insufficently similar to physiological and also valid.

There are a number of gramatical errors relating to the use of articles (e.g. line 258 'The further subdivision is based on the type of protein and lipid conjugate' should read 'A further subdivision...') and number  and inappropriate use of adjectives (e.g. line 222 'at specialize cellular localizations' should be either 'at specialized cellular locations' or possibly 'at specific cellular locations')

 Some typographical and spelling errors. e.g line 243 'brunched biopolymers' and in the legend for figure 2, lane is repeatedly misspelt lain.

Motif is often misspelt motive. 

Line 809 'decompaction will occur only in the presence of trivalent Lex and not trivalent Le.' What does this mean? 

Reviewer 2 Report

The review paper describes glycan to glycan binding, as a novel type of interaction applicable in nature.

In my opinion, the review paper is worth publishing after major revision by addressing the following comments/questions:

  1. Define the homophilic and heterophilic type of interaction (line 135).
  2. Can you describe details about the synthesis of glycans extending several hundreds of nanometers from the plasma membrane in a chapter/sub-chapter? How glycosyltransferases operate so far away from the membrane surface?
  3. In my opinion, a larger number of tripeptides than just 6 are possible using 3 amino acids (line 310) i.e. AAA, AAB, ABA, AAC, ACA, ABC, ACB, AAD, etc. Please revise it accordingly.
  4. Define terms "autologous, allogeneic, xenogeneic heterogeneity" (line 317).
  5. Authors need to provide references for that statement (line 331-335).
  6. Is there any review paper describing glycan to glycan recognition? If yes, how is this review different from other review papers?
  7. The authors need to list other methods besides AFM i.e. describe what optical methods and other methods to study binding glycan to glycan affinity (lines 343-344).
  8. There is a very low informative value in Figure 2. It is not possible to see what is approximate molecular weight by showing PAGE with some standard.
  9. Replace "Lain" with "Lane" in the figure caption to Fig. 2.
  10. There is no lane d and lane e shown in Figure 2B. Please provide it.
  11. What kind of useful information is shown in Fig. 2C?
  12. What is the molecular mass of hydrolysate shown in Fig. 2E?
  13. Fig. 3 with higher quality/resolution should be provided.
  14. Definition of carbohydrates containing trisaccharide on lines 432 and 433 need to be provided (line 432).
  15. What kind of results indicates the functional role of these glycan sequences in cell adhesion (lines 436-437)?
  16. How is the selectivity of glycan-glycan interaction achieved when Ca2+ is needed for such interactions? Why is  glyconectin x not binding to glyconectin y, but rather glyconectin x is bound to glyconectin x and glyconectin y binds preferentially to glyconectin y?
  17. Why physiological concentration is needed for glycan to glycan binding? It is easily understandable why in the absence of Ca2+ such interaction does not occur, but why it does not occur if Ca2+ concentration is higher than physiological?
  18. Is the binding force properly described as 30 nm (lines 690-691)?
  19. Provide some SPR data in a form of Figures, when desribinf interaction of Lex with Lex.
  20. Replace "Lex" with "Lex" (lines 882, 904,...)
  21. Conclusions are needed and the section describing the challenges to be addressed in the future, as well.

Reviewer 3 Report

In this review article entitled “Glycan to Glycan Binding: Molecular Recognition Through Polyvalent Interactions Mediates Specific Cell Adhesion”, Misevic and Garbarino discussed molecular interactions between glycans and glycans which regulate cell adhesion in the specific biological contexts. Initially, the authors describe Glycan to glycan molecular interactions as biologically relevant biopolymeric interactions. Then, the authors present the rationale for glycan to glycan binding concept in cellular interactions with the information about the topology, abundancy, and spatiotemporal control of glycans on cellular membranes. Finally, the authors describe two examples, glyconectin and Lex, as molecules that show glycan to glycan binding.

This is a summary of the very interesting subject of glycan-glycan interactions. It will be of interest to many researchers.

I would ask the authors to address my specific concerns below.

(1) There are no major problems with the content or framework of the article. However, the context of paragraphs 899 to 901 is confusing. It may be better to omit it. If left in, it would be necessary to make the context easier to grasp and to cite references.

(2) The entire text including the figure legends needs to be revised carefully, as there are many grammatical errors, spelling errors, and missing spaces. For example, in line 21 of the abstract, “relay on” should be “rely on”. Also, the authors use a lot of “motive”, but they should use “motif” for all of them; they both write AFM and spell it out. Only the first one should be spelled out, and the rest should be AFM. The way Lex is written, with or without superscript, should also be standardized. In the legend of Figure 5, line 471, neroglycolipids is a word I have never heard.

(3) In line 436, the sentence beginning with "These results indicate~" has no verb. If the word "indicate" is to be used, it should be toned down and replaced with "strongly suggest" because the fact that the epitope of the inhibitory antibody was a sugar does not mean that the sugar is the molecule involved in adhesion itself.

(4) What does AMBER mean in line 848?  It should be spelled out.

Round 2

Reviewer 2 Report

The authors did a substantial revision of the manuscript addressing almost all my comments and questions. However, there is still a comment not properly addressed.

The authors need to cite previous review papers dealing with glycan to glycan recognition with a short description of what can be found in those already published review papers. This could be of interest to the readers of this review paper in case they would like to get more information about glycan to glycan recognition in the areas, which are not covered in this review paper.